# Portosystemic shunt placement reveals blood signatures for the development of hepatic encephalopathy through mass spectrometry

Ana Carolina Dantas Machado[1,20], Stephany Flores Ramos [iD][1,2,3,20], Julia M. Gauglitz[4], Anne-Marie Fassler[5], Daniel Petras [iD][4,6], Alexander A. Aksenov[4,7], Un Bi Kim[5], Michael Lazarowicz[8], Abbey Barnard Giustini[1,9,10], Hamed Aryafar[11,12], Irine Vodkin[1], Curtis Warren[5], Pieter C. Dorrestein [iD][4,13,14,15], Ali Zarrinpar[5,16,17] ✉ & Amir Zarrinpar [iD][1,13,18,19] ✉

Elective transjugular intrahepatic portosystemic shunt (TIPS) placement can worsen cognitive dysfunction in hepatic encephalopathy (HE) patients due to toxins, including possible microbial metabolites, entering the systemic circulation. We conducted untargeted metabolomics on a prospective cohort of 22 patients with cirrhosis undergoing elective TIPS placement and followed them up to one year post TIPS for HE development. Here we suggest that pre-existing intrahepatic shunting predicts HE severity post-TIPS. Bile acid levels decrease in the peripheral vein post-TIPS, and the abundances of three specific conjugated di- and tri-hydroxylated bile acids are inversely correlated with HE grade. Bilirubins and glycerophosphocholines undergo chemical modifications pre- to post-TIPS and based on HE grade. Our results suggest that TIPS-induced metabolome changes can impact HE development, and that pre-existing intrahepatic shunting could be used to predict HE severity post-TIPS.

Hepatic encephalopathy (HE) is a common complication of advanced liver disease, with up to 30–40% of patients developing this neurological condition as their disease progresses[1]. The phenotype of HE can vary widely from mild cognitive impairment to more serious symptoms such as disorientation and coma. While the etiology of HE is complex, the most prevalent theory of its pathophysiology is the inability of dysfunctional hepatocytes to process enteric neurotoxins that accumulate within the splanchnic circulation, allowing them to enter the systemic circulation, and precipitate cerebral inflammation and cerebral edema[2,3].

Due to the elevated pressure within the portal venous system and resultant hepatofugal flow in the cirrhotic liver, extrahepatic collateral blood vessels known as varices form to allow for portal blood return to the vena cavae and the systemic circulation[4]. Porto-hepatic venous collaterals, or intrahepatic shunts, can also form, although efforts to

characterize this phenomenon have been sparse[5–7]. The presence of these extrahepatic and intrahepatic shunts alters toxin circulation, decreases toxin metabolism in the liver, and contributes to hepatic encephalopathy[8,9].

In severe liver disease with refractory ascites or variceal bleeding, transjugular intrahepatic portosystemic shunt (TIPS) placement is a therapeutic intervention that can reduce pathologies caused by portal hypertension. With TIPS, a synthetic shunt connects the portal venous system to the hepatic venous system, such that some blood can bypass the cirrhotic liver and reduce portal hypertension and its sequelae. Although TIPS can significantly improve the morbidity and mortality of patients with cirrhosis and increase transplant-free survival, TIPS placement is also a known risk factor for the development of HE and, thus, preexisting HE is a relative contraindication to TIPS placement. TIPS can increase the incidence of HE by 30–55%

post-procedure, with 90% of affected patients developing it within the first three months[10–13].

While the association between TIPS placement and HE exacerbation is well-documented, the mechanism is poorly understood. Ammonia level is usually associated with HE, but studies examining ammonia level and HE development are conflicting. The association is highly dependent on the chronicity of hyperammonemia and varies greatly in outcome and thus levels are not typically followed clinically[14–17]. Therefore, neurotoxic compounds beyond ammonia likely contribute to HE. Microbial products such as aromatic amino acids and bile acids (BAs) are candidate neurotoxins, but it is still unclear which microbial products or metabolites induce HE onset and how TIPS placement affects the concentration of these compounds within the systemic circulation[3,18–20].

A promising method to identify such metabolites is untargeted metabolomics, which has been extensively applied to study microbial community function and host-microbe interactions[21–26]. Targeted metabolomics allows for absolute quantification of specific metabolites such as amino acids and fatty acids and is limited to a few previously identified compounds for which authentic standards are available. Untargeted metabolomics, on the other hand, allows for measurement of both known and unknown metabolites, thus enabling the discovery of novel metabolites and insights into biological function[23,27]. A key recent advance in untargeted metabolomics is the development of ion identity molecular networking (IIMN) to expand the annotation and quantification of metabolite features by enabling feature correlation analysis coupled with matching among spectral libraries and molecular networking[28–33]. Even for features that cannot be fully annotated, molecular networking can be used to infer compound subclass level annotation (e.g., bilirubins, bile acids) through the network connections. The potential breadth of information that can therefore be gained from untargeted metabolomics with these recently developed bioinformatic tools can lead to the identification of novel therapeutics and biomarkers[24,34–38]. Moreover, by obtaining serial samples from a single subject, investigators can prioritize proportional changes between biosynthetically related compounds, and postulate putative biotransformations, and determine what effects these changes may have on overall physiology[39].

In this study, we performed an untargeted metabolomic analysis on hepatic vein and peripheral vein blood samples from patients undergoing TIPS. Our primary outcome was to understand which metabolite levels change as a result of portosystemic shunt placement and our secondary outcome was to determine whether these changes can predict worsening HE. To do so, we used standard and innovative untargeted metabolomics analyses that increased the overall set of annotations and allowed an exploration of the potential chemical modifications undergone by, and changes in levels of, the compounds after portosystemic shunt placement. We hypothesize that early shifts in these compounds predict the development and severity of HE and suggest the presence of intrahepatic shunting in some cirrhotic livers that is predictive of future development of HE.

## Results

### Characterization of metabolites before and after TIPS placement implicates plasma bile acids levels

A total of 22 participants underwent TIPS placement at one of the two centers (Supplementary Fig. 1). Just prior to TIPS placement, 20 participants provided a peripheral vein sample (Fig. 1a). During the TIPS procedure, the interventional radiologist collected hepatic vein blood for 20 participants just prior to and just after shunt placement. After the procedure but prior to admission to the post-anesthesia care unit, 20 of the participants provided another peripheral vein blood sample. Finally, 20 of the participants provided a fasting peripheral vein blood sample on the day of discharge with their morning labs (i.e., before discharge) (Fig. 1a). In total, 19 individuals had both pre- and post peripheral vein, and 18 individuals had pre- and post-peripheral and hepatic vein samples, thus allowing paired analysis for many of our comparisons. It was important to collect samples from both the peripheral and hepatic vein because the hepatic vein samples provided information on changes occurring immediately after blood flowed through the liver, while the peripheral vein samples, collected through an easier method, revealed systemic changes that can be used for diagnosis and treatment. Participants were monitored by chart review for up to a year after their procedure and the worst HE grade during that period for each participant was noted. For two of the individuals who were readmitted with HE exacerbation during the study period, additional peripheral vein blood samples were also obtained. Participant demographics and clinical characteristics are described in Supplementary Table 1.

We interrogated differences in molecular distributions revealed by untargeted metabolomics associated with TIPS and HE severity. To do so, we characterized the metabolomic composition of blood plasma samples collected directly from hepatic and peripheral veins pre- and post-TIPS, as well as before discharge. Metabolite features identified through untargeted LC/MS-MS were clustered by MS/MS similarity and annotated with a spectral library using feature-based molecular networking (FBMN) (Supplementary Fig. 2a) through the Global Natural Products Social Molecular Networking (GNPS) platform [https://gnps.ucsd.edu/][29]. This resulted in the annotation of 234 features (39.3%), referred to as metabolites, among several clusters within the network (Supplementary Fig. 2b, c). This annotation rate, which greatly exceeds typical annotations (below 10%), is enabled through the use of the Nearest Neighbor Suspect Spectral Library[40].

Using the untargeted metabolomic data, we investigated how TIPS placement affects the metabolome in the peripheral and hepatic vein samples (Fig. 1b–d). Metabolome dissimilarities across participants (as measured using the robust Aitchison β-diversity) pre- to post-hepatic vein were not significantly different (PERMANOVA $P = 0.89$, Fig. 1b), but the difference reached significance between pre- and post-peripheral vein samples (PERMANOVA $P = 0.021$, Fig. 1c). To account for post-TIPS changes that might take longer to manifest, we assessed the dissimilarity of the samples taken before discharge to our earlier peripheral vein samples and observed no change pre-peripheral vein to before discharge (PERMANOVA $P = 0.149$) but a strong change post-peripheral vein to before discharge (PERMANOVA $P = 0.003$, Fig. 1c). These results indicate that the most substantial metabolomic changes in our samples did not occur immediately after TIPS placement but rather after TIPS placement and before discharge.

Since the most significant metabolome changes occurred between the post-peripheral vein and before discharge metabolome, we interrogated which metabolites are driving these shifts. To do so, we used Qurro to visualize the log-fold change (rank) of metabolites contributing to this difference[41]. This analysis demonstrated that bile acids, as a group, were major contributors to the dissimilarity (Supplementary Fig. 3). To better quantify this shift in bile acids, we used the natural log ratio of bile acids to bilirubins, a group of metabolites that were similarly present in all three peripheral vein timepoints (Supplementary Fig. 3). This analysis demonstrated that the abundance of bile acids drops immediately after shunt placement, but their abundances are restored to pre-peripheral vein level by the time of discharge (Fig. 1d, Supplementary Fig. 3).

Bile acid changes across different time points along the TIPS procedure are also highlighted in paired Wilcoxon analyses of individual metabolites. Twenty-four metabolites were significantly different pre- vs. post-peripheral vein while 12 were different pre- vs. post-hepatic vein. (FDR < 0.1; Fig. 1e). These two sets of metabolites overlap heavily; together, 25 unique metabolites are significantly different between the pre- to post-TIPS samples. The only differentially abundant metabolite with an annotation in the pre- to post-peripheral vein analysis was glycoursodeoxycholic acid (GUDCA), which decreases in

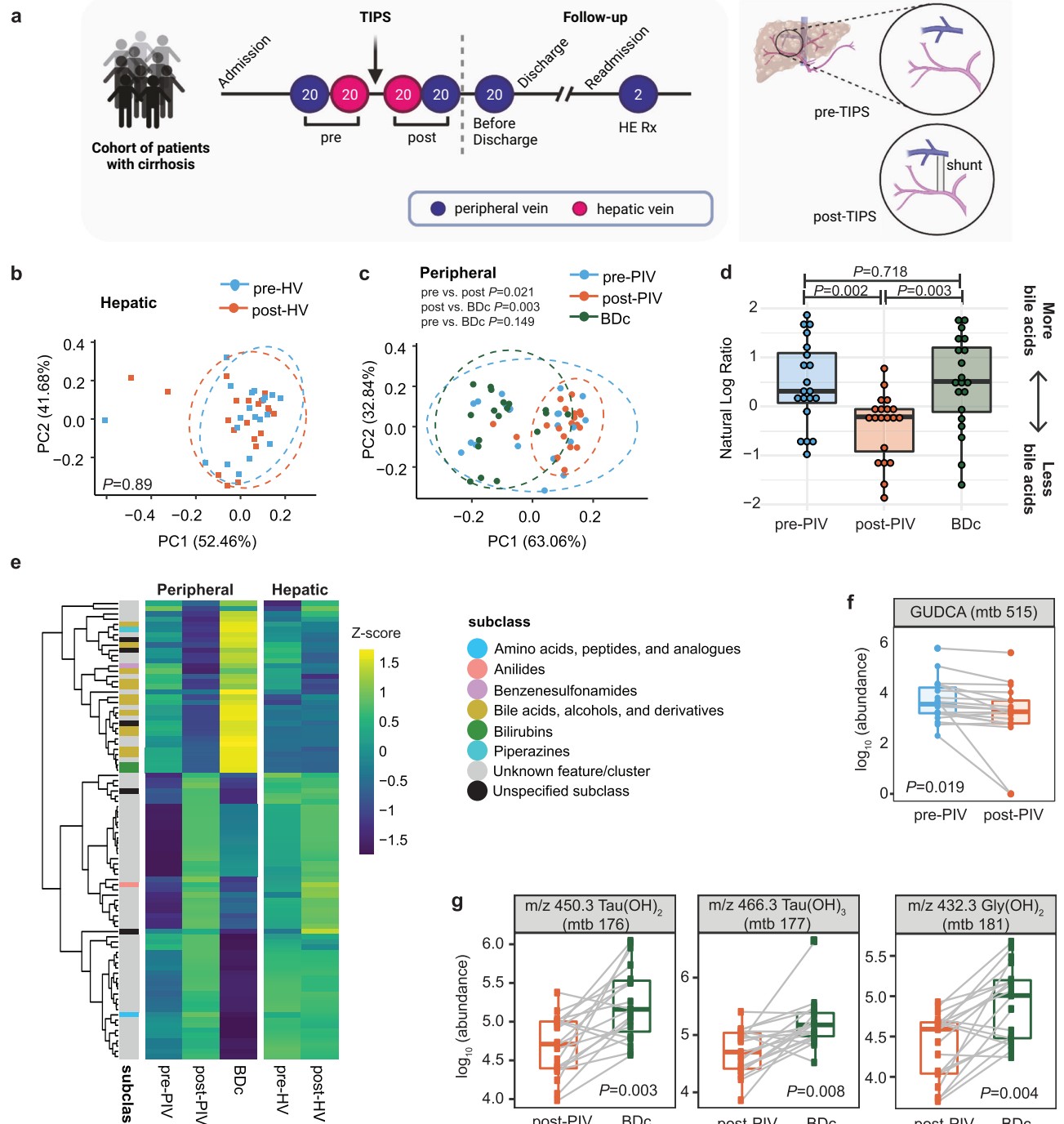

**Fig. 1 | The effect of TIPS placement on the hepatic and peripheral metabolomes. a** Blood sample collection design. Peripheral vein (purple) and hepatic vein (pink) blood samples were collected pre- and post-TIPS, and following readmission for HE treatment. Visual of the location a shunt is placed during TIPS. Created with Biorender.com. **b, c** Robust principal component analysis (RPCA) showing metabolome dissimilarities across participants in (**b**) pre-hepatic vein (sky blue squares) and post-hepatic vein (orange squares) samples and (**c**) pre-peripheral vein (sky blue circles), post-peripheral vein (orange circles), and before discharge (green circles) samples. **d** The natural log ratio of bile acids to bilirubins (pairwise Wilcoxon, FDR corrected at q-value 0.05, $n = 20$ participants). **e** Heatmap showing the 87 unique metabolites that were significantly different between at least two of the three timepoints (paired two-sided Wilcoxon, FDR corrected at q-value 0.1); 25 unique metabolites significantly different pre vs. post-TIPS and 84 significantly

different post-peripheral vein vs. before discharge (FDR < 0.1). **f** Pre- vs. post-peripheral vein levels of GUDCA, (paired two-sided Wilcoxon $P = 0.019$, $n = 19$ participants). **g** Examples of bile acids that were significantly different post-peripheral vein vs. before discharge (paired two-sided Wilcoxon, FDR corrected at q-value 0.05, $n = 19$ participants). (GUDCA=Glycoursodeoxycholic acid; m/z 450.3 Tau(OH)$_2$ (mtb 176), m/z 466.3 Tau(OH)$_3$ (mtb 177), and m/z 432.3 Gly(OH)$_2$ (mtb 181) are annotated as bile acids through suspect library matching). The boxplots show median and upper and lower quartiles. The extreme lines show the highest and lowest value. The boxplot is overlaid with the visualization of single observations. Colors represent samples collected at different time points: sky blue (pre-TIPS); orange (post-TIPS); green (before discharge). PIV=peripheral vein; HV = hepatic vein; BDc = before discharge. Source data are provided as a Source Data file.

the peripheral plasma immediately following the TIPS procedure (Fig. 1f). Meanwhile, 84 metabolites were differentially abundant post-peripheral vein to before discharge, with the majority mapping to annotated bile acids (Fig. 1g). Taken together, these results demonstrate that TIPS placement is associated with changes in the abundance of bile acids in peripheral circulation.

### Post-TIPS HE severity is higher in participants with potentially less intrahepatic shunting

The surprisingly small difference observed between the pre- and post-hepatic vein metabolome strongly indicates considerable intrahepatic shunting is already occurring in most participants prior to undergoing TIPS. The shunting is necessarily intrahepatic, because the blood being sampled comes directly from the hepatic vein and thus would not include other portosystemic shunts such as esophageal varices or splenorenal shunts.

To explore how this could affect the future development of HE, we analyzed metabolome changes based on the worst HE grade within a year after the TIPS procedure. Metabolome dissimilarities across participants post-TIPS for peripheral vein and hepatic vein show no significant differences for any of the pairwise HE grade comparisons (PERMANOVA all pairwise $P > 0.05$), although the spread of points increases with increasing severity for just the hepatic vein samples (Fig. 2a, Supplementary Fig. 4a). To ensure that this relationship between HE grade and variability was not due to reduction in portal hypertension after TIPS, we compared HE grade and changes in direct portal vein pressure and did not see a significant association (Fig. 2b).

We then hypothesized that participants who develop more severe HE have less intrahepatic shunting at the time of their TIPS. To test this hypothesis, we compared the shift in metabolome within each participant before and after the TIPS procedure. In this analysis, those who already have significant intrahepatic shunting should have a low dissimilarity between their pre- and post-TIPS hepatic vein plasma metabolome, while those who have less shunting should have a higher dissimilarity. Within-participant dissimilarities between pre- and post-peripheral vein metabolome did not show any difference based on HE grade (Fig. 2c). Serendipitously, within-participant dissimilarities between pre- and post-hepatic vein metabolome are significantly different between participants with an HE grade of 0 versus participants with a grade of 1 or 2+ (Wilcoxon $P = 0.027$ and $P = 0.036$, respectively, Fig. 2d), despite no overall difference in hepatic vein blood metabolome before and after TIPS (Fig. 1b). This difference is also consistent when we remove HE grade 1 (Supplementary Fig. 4b, c). This change in the hepatic vein metabolome based on HE grade suggests that individuals with more dissimilarity in their metabolome after shunt placement, likely from a lack of pre-TIPS intrahepatic shunting, are at a higher risk of developing HE after the procedure (Fig. 2e).

### More severe HE outcomes are associated with decreased post-shunt levels of bile acids and glycerophosphocholine

Our analyses indicate that participants who developed HE had increased dissimilarity in their hepatic vein metabolome between their pre- and post-TIPS samples (Fig. 2d). To determine which metabolites are related to later development of HE, we analyzed how individual metabolites changed from baseline pre-TIPS levels (also called "change from baseline" and defined as log10(abundance$_{post}$/abundance$_{pre}$) for peripheral vein or hepatic vein samples. Assessing both sample types separately allows us to focus directly on the hepatic vein, where most changes resulting from TIPS are likely occurring, and also identify metabolites in the peripheral blood, which is more easily accessible and diagnostically valuable. This analysis revealed that the change in individual metabolite levels in participants who develop post-TIPS HE (grade 1 or 2+) is quite different from that of participants with no symptoms of HE (grade 0, Fig. 3a, b). Overall, participants with more severe HE (2+ grade) exhibit a different distribution of the change from

baseline levels of their metabolites (Fig. 3b). This is observed for both hepatic and peripheral samples and characterized by an increased number of metabolites exhibiting lower change from baseline for participants who develop severe HE.

Out of 64 clusters (groups of related metabolites, Supplementary Fig. 1), two clusters with annotated compounds had significant differences based on HE severity: bile acids and glycerophosphocholines (Fig. 3c, d). Larger decreases from baseline in bile acids were observed for participants with more severe HE in peripheral and hepatic vein samples, while a similar trend for glycerophosphocholine was observed in peripheral plasma (Fig. 3c, d). Changes from baseline levels of additional clusters were significantly different between HE grades but were not associated with HE severity. Consequently, decrease in bile acids (in hepatic vein and peripheral vein) based on their baseline levels is a characteristic of patients who undergo TIPS and later develop HE and could be used to determine patients at risk of developing HE post-TIPS.

### Low levels of post-shunt bile acids are associated with increased hepatic encephalopathy severity

Our results thus far demonstrate that bile acids exhibit differences from baseline levels that are related to the development of HE. This suggests that these metabolites play a role in HE pathophysiology (Fig. 3), and their detection in peripheral vein, which is much more accessible, may help distinguish patients who may be at a high risk for post-TIPS HE. A closer examination of the bile acid metabolome showed that the abundances of three conjugated di- and tri- hydroxylated bile acids in the post-peripheral vein are significantly correlated with HE grade. All three decrease with higher HE grade, suggesting that circulating levels of these bile acids post-shunt placement are inversely associated with HE severity (Fig. 4a). Differences in bile acids based on HE grade remain consistent when removing HE grade 1 or grouping the two covert HE grades (0 and 1) together (Supplementary Fig. 5).

We next analyzed longitudinal bile acid levels for two participants (participants 11_a and 12_a) who were readmitted to the hospital within one year for HE treatment (HET) following a TIPS procedure. We investigated whether these three particular bile acid levels were also impacted at the time of readmission and treatment for HE (Fig. 4b, c). While these results are representative of only two participants, it is noteworthy that there is a drop in the peripheral vein levels of these bile acids immediately after shunt placement, in agreement with the results based on their later HE grade (Fig. 4a–c). Interestingly, upon readmission due to HE symptoms, bile acid levels are lower for participant 11_a, and HE treatment increases bile acid levels compared to pre-HET (Fig. 4b). An increase in these bile acids post-HET following readmission is also demonstrated for participant 12_a (Fig. 4c). The clinical course, demographics, and etiology for each readmitted participant revealed no overt predisposing clinical factors to explain the difference in the participants' bile acid levels. Thus, these hydroxylated conjugated bile acids, which we suspect are secondary bile acids, may play a role in HE pathophysiology or predict which patients will develop more severe HE.

### Abundance and mass shifts in glycerophosphocholines and bilirubins pre- to post-TIPS suggest potential hepatic bio-transformations that prevent hepatic encephalopathy

The previous analyses demonstrated that TIPS placement disrupts the levels of specific metabolites in participants undergoing the TIPS procedure (Fig. 1c, e, f), and that these changes are related to HE grade (Figs. 3c, d and 4a). We next aimed to investigate the impact of shunt placement on the dynamic chemical transformations of related metabolites and the underlying changes in liver metabolic pathways. To this end, we used an approach called chemical proportionality (ChemProp), which takes advantage of longitudinal metabolite

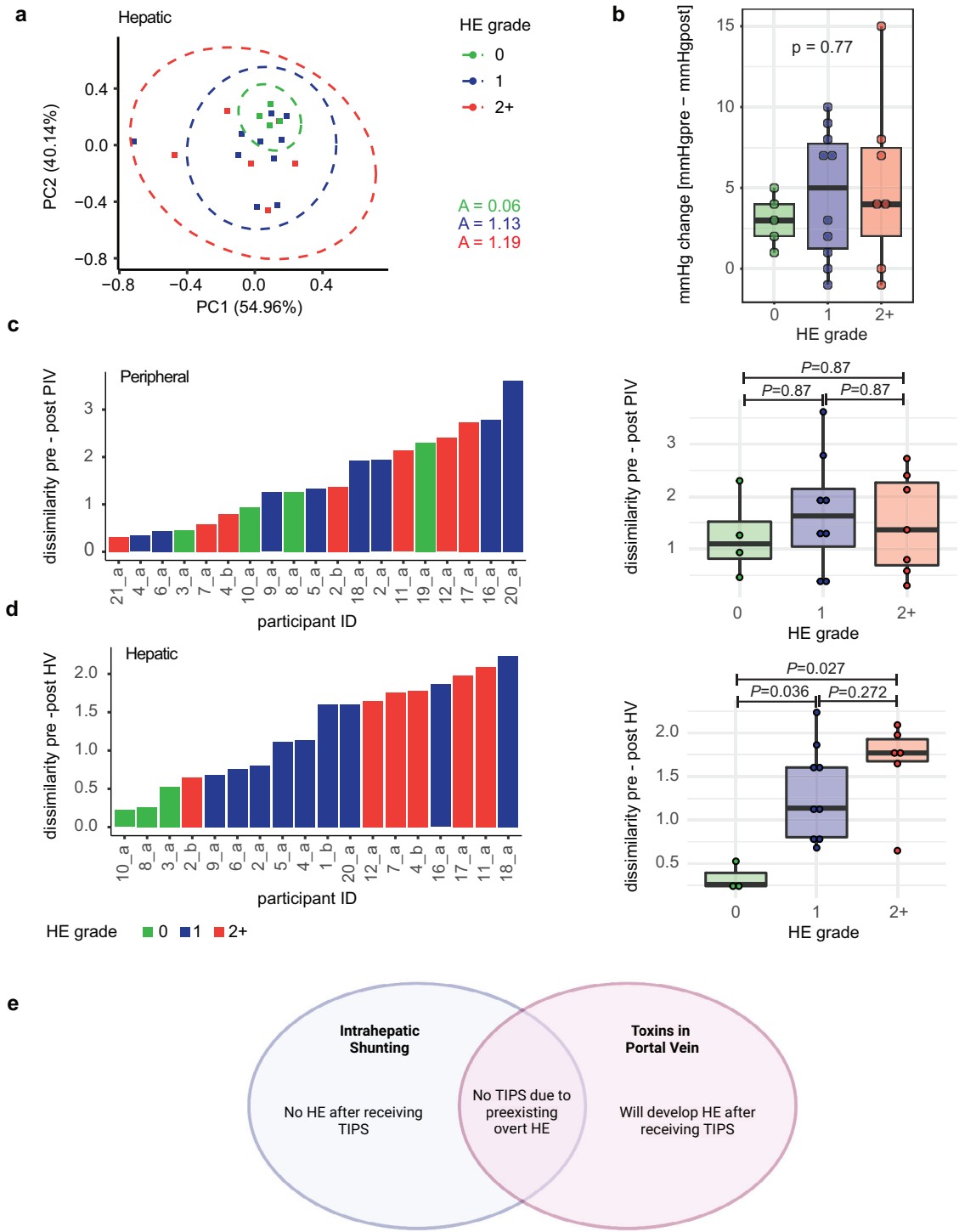

**Fig. 2 | TIPS placement effect on metabolite patterns based on HE severity.**
**a** RPCA plot showing metabolome dissimilarities across participants post-TIPS in hepatic vein samples stratified by HE grade. A is the area within the ellipsis.
**b** Pressure change was calculated as the difference between direct portal vein pressure measurements pre- and post-TIPS (Kruskal–Wallis test, $n = 22$ participants). Comparison of the (**c**) peripheral vein ($n = 19$ participants) and (**d**) hepatic vein ($n = 18$ participants) dissimilarity distances pre- to post-TIPS for each participant based on HE grade. Left: a ranked bar plot showing the (**c**) pre- to post-peripheral vein or (**d**) pre- to post-hepatic vein dissimilarity distances for each

participant. Right: boxplot showing the same data but grouped by participant HE grade (pairwise Wilcoxon, FDR corrected at $q$-value 0.05). Dissimilarity is defined as the robust Aitchison β-diversity calculated with DEICODE. HE grade: 0 = none (green); 1 = mild (blue); 2+ = severe (red). The boxplots show median and upper and lower quartiles. The extreme lines show the highest and lowest value. The boxplot is overlaid with the visualization of single observations. **e** Theoretical framework on the relationship between intrahepatic shunting, portal neurotoxins, and development of HE. Created with Biorender.com. PIV peripheral vein; HV hepatic vein; BDc before discharge. Source data are provided as a Source Data file.

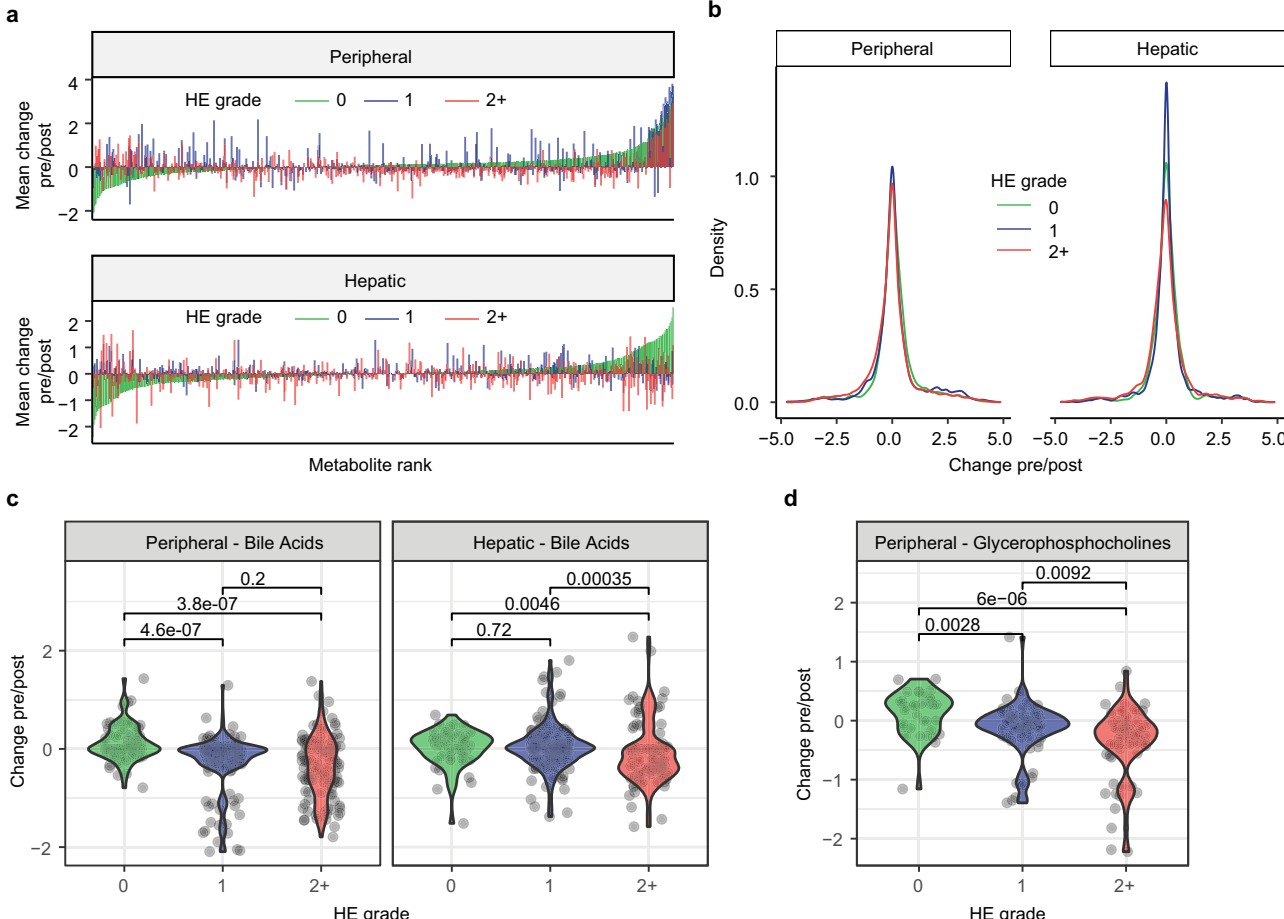

**Fig. 3 | Change from baseline (pre/post) for individual metabolites. a** Mean change in pre/post abundance for each metabolite based on HE grade for peripheral or hepatic vein samples. Features are ordered in ascending order for HE grade "0" group. **b** Density plot of the change pre/post in metabolite abundances of each participant based on HE grade. Kolmogorov–Smirnov test peripheral: 0 vs 2+, $P = 1.3e-13$; 0 vs 1, $P = 1.0e-03$; 1 vs 2+, $P = 2.4e-09$; hepatic: 0 vs 2+, $P = 5.7e-08$; 0 vs 1, $P = 0.08$; 1 vs 2+, $P = 2.2e-16$. Change pre/post TIPS for bile acids (peripheral and hepatic vein samples) (**c**) and glycerophosphocholines (peripheral samples) (**d**) between participants based on their HE grade (0, 1, or 2+). Change pre/post is defined as the log10(abundance$_{post}$/abundance$_{pre}$) for each participant's metabolites. HE grade: 0 = none (green); 1 = mild (blue); 2+ = severe (red). Source data are provided as a Source Data file.

abundance data associated with FBMN and calculates a ChemProp score for each metabolite pair (Fig. 5a, b)[39]. In our study, a high ChemProp score helps us identify the main dynamic changes occurring in individual metabolite pairs pre- to post-TIPS based on their abundances. At the same time, inspection of mass differences (delta m/z) between the same metabolites helps us identify potential biotransformations that may be responsible for the changes in their abundance (Fig. 5b). As a result, this approach allows us to prioritize putative functions in the liver that are most likely to be affected by TIPS placement and to gain a deeper understanding of the underlying metabolic pathways associated with it (Fig. 5b).

The principal changes identified through ChemProp in the hepatic vein are related to metabolite pairs within the glycerophosphocholine and bilirubin clusters, indicating that these metabolites fluctuate to a greater degree pre- to post-TIPS (Fig. 5c). For example, the mass change of 24.00 (m/z) and a high ChemProp score of 1.3 between two closely related glycerophosphocholines suggests that significant fluctuations in the number of carbons (C2) occur before and after TIPS (Fig. 5d). We also observe a high degree of fluctuation between two compounds within the bilirubin class, with a mass change of 2.021 (m/z) and a ChemProp score of 1.06, where one of the compounds matches biliverdin (Fig. 5e). These results suggest that TIPS placement greatly impacts biliverdin reduction in the hepatic vein, possibly due to an increase in the reduced form post-TIPS (Fig. 5e).

When taking HE grade into account, we additionally identified specific sets of metabolites fluctuating at different degrees pre- to post-TIPS in patients developing HE (Fig. 5f). For example, biliverdin oxidation might play a role in post-TIPS HE, as there is a high Chem-Prop score between biliverdin and another metabolite with a potential delta m/z of 16.031 in patients with HE grade 2+ (Fig. 5g). Additionally, a mass shift of 23.99 (C2) between two glycerophosphocholines is high in patients with severe HE (Fig. 5h). Interestingly, most bile acid ChemProp scores are low, suggesting that the TIPS procedure itself is not altering the ratios of bile acid transformations pre- to post-TIPS (Supplementary Fig. 6). This is consistent with the idea that most of these bile acid transformations are occurring in the gut and not in the liver.

## Discussion

HE is a serious complication of liver disease, and TIPS placement can increase its risk. The etiology and treatment of HE have historically focused on ammonia homeostasis; however, ammonia levels are not universally predictive of the degree of HE and are not reliable in guiding treatment[42,43]. In light of this incongruence, additional neurotoxins have been implicated, but no clinically applicable biomarker for HE severity has been validated. In this study, we used the latest methods in untargeted metabolomics with an expanded set of annotations to characterize metabolites implicated in changes post-TIPS

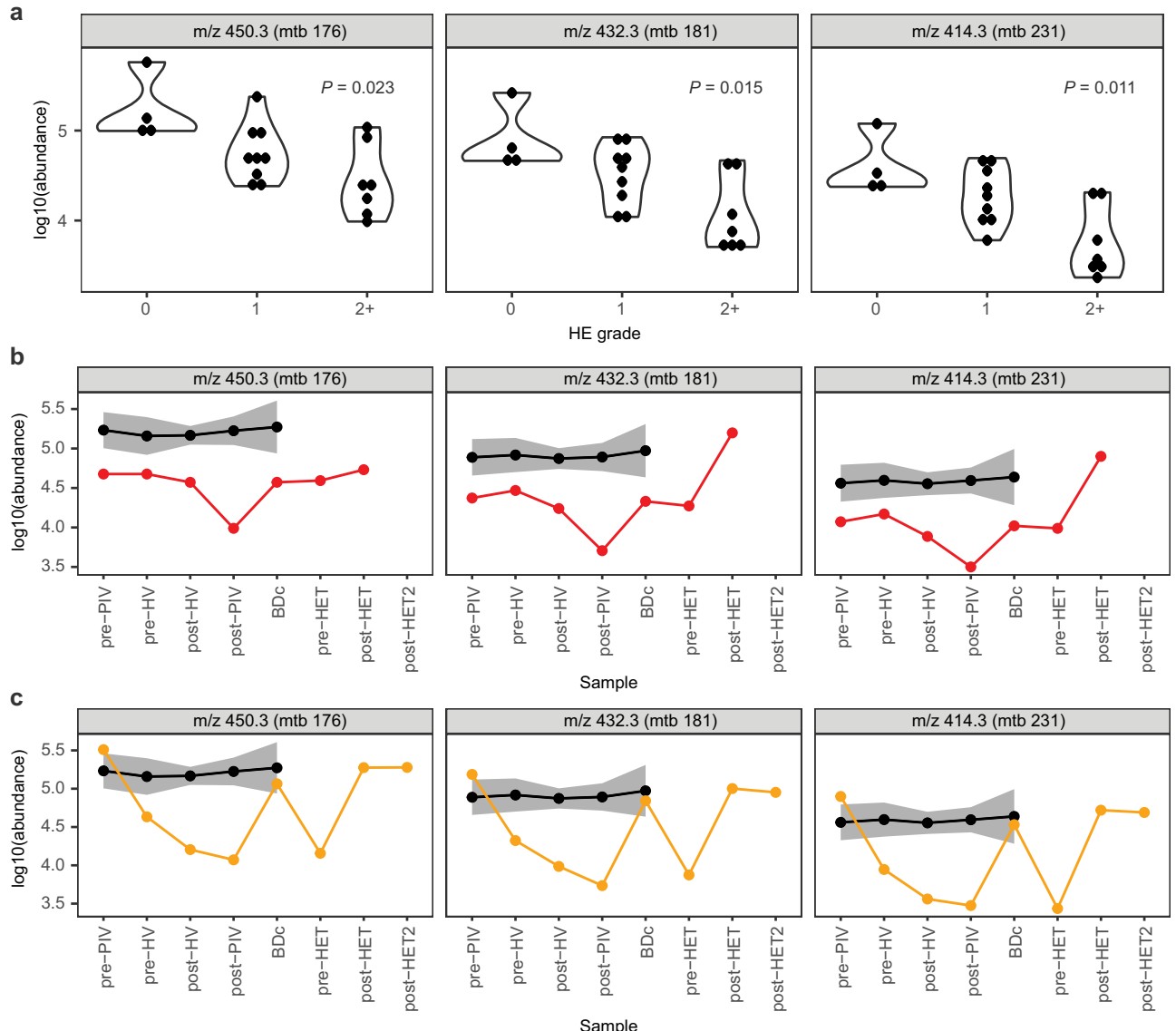

**Fig. 4 | Levels of bile acids in plasma of TIPS participants based on HE grade.**
**a** Bile acid levels and significant abundance differences in the post-TIPS peripheral blood based on HE grade. HE grade: 0=none; 1=mild; 2 + =severe (Kruskal-Wallis test, FDR < 0.2). Longitudinal levels of bile acids in two participants readmitted with HE grade 2+ represented by red (**b**, participant 11_a) and orange (**c**, participant 12_a) compared to participants with HE grade 0 (black line, *n* = 4; shaded areas represent SEM). PIV peripheral vein, HV hepatic vein, BDc before discharge, HET hepatic encephalopathy treatment. Source data are provided as a Source Data file.

that could lead to the development of HE. With further study, these compounds may be used as biomarkers and potential therapeutic targets for HE. Notably, the methods described herein are not limited to HE; they can be applied in virtually any disease process to identify involved metabolites, which can be investigated as biomarkers or targets for drug discovery.

Untargeted metabolomics allows us to compare metabolites within the peripheral and hepatic circulations before and after TIPS placement. Both peripheral and hepatic veins are important to our analysis. Whereas samples from the hepatic vein, which is much harder to access in patients, inform us of intrahepatic shunting and biochemical transformations that occur in the liver (through our Chem-Prop analysis), the peripheral vein samples, which are much easier to obtain, can help us determine whether specific metabolomic findings can be used as a potential biomarker of disease activity. In our study, we observed that the metabolome pre- to post-TIPS changes to a greater extent in the peripheral vein compared to the hepatic vein (Fig. 1b, c), and the metabolites contributing the most to these changes

are bile acids (Fig. 1d–g and Supplementary Fig. 3). The surprisingly small differences in hepatic vein blood metabolome before and after TIPS suggests that the overall blood metabolome does not drastically change as a result of TIPS. However, participants with the greatest degree of dissimilarity in the pre- to post-TIPS hepatic vein metabolome went on to develop HE (Fig. 2d). The higher degree of metabolome dissimilarity in HE grades 1 and 2+ implicates a functional and perhaps structural difference in participants who developed HE, namely, intrahepatic shunting.

The phenomenon of intrahepatic shunting has been described in the previous literature. Specifically, this process is synonymous with pathologic hepatic angiogenesis, which is provoked by chronic liver disease[6,44,45]. The microenvironment of inflammation, oxidative stress, shear stress, and hypoxia results in the production of angiogenic and endothelial growth factors[46]. Vascular remodeling or "capillarization" occurs within hepatic sinusoids, leading to altered hepatocyte blood flow. This can diminish the synthetic and detoxifying mechanisms within the previously functioning hepatocytes, as they are bypassed by

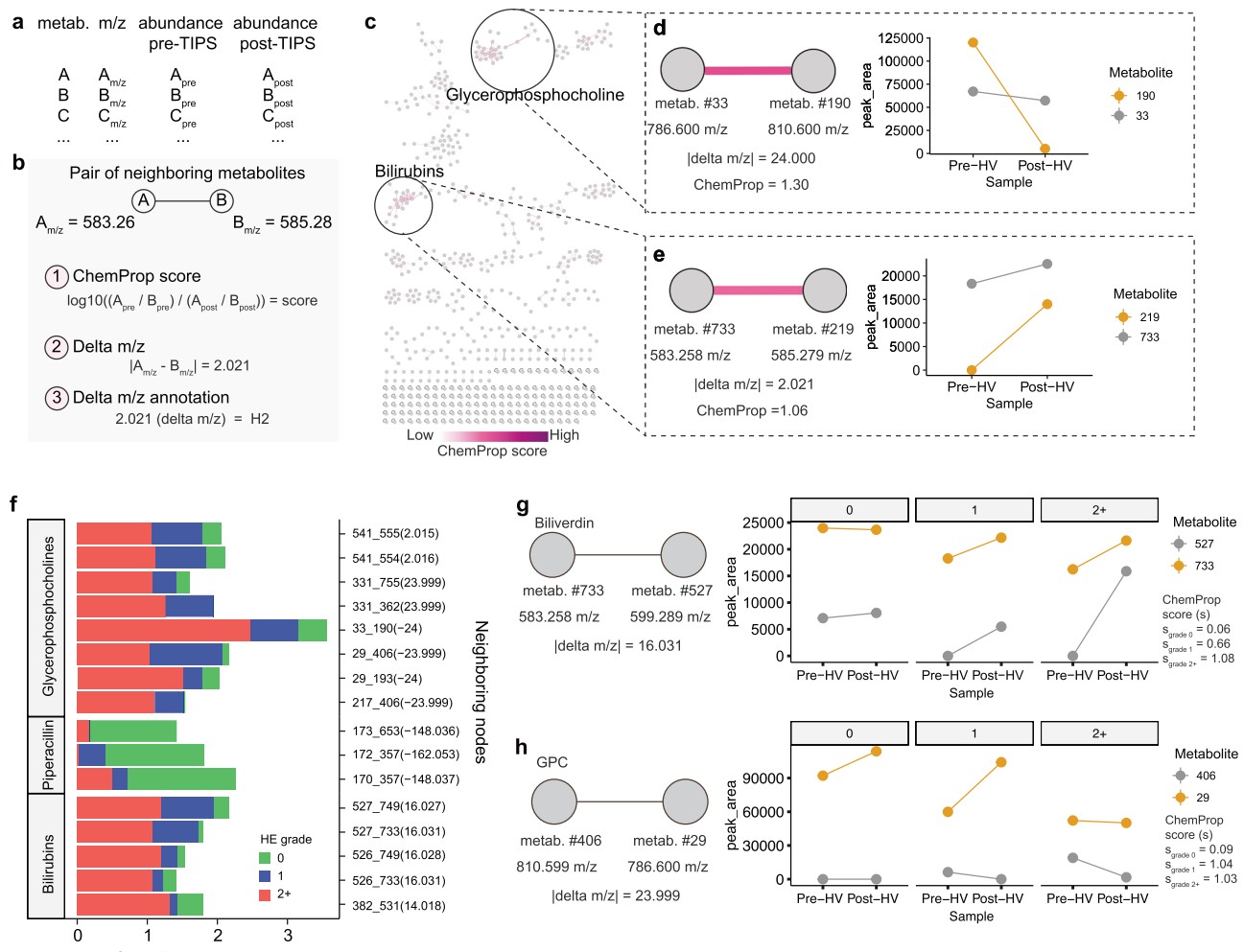

**Fig. 5 | Chemical proportionality of neighboring metabolites pre- to post-TIPS placement. a, b** Schematic representation of the (**a**) quantification table from FBMN and (**b**) data that can be deduced from neighboring metabolites, including the ChemProp score, which is calculated as the log-ratio of two neighboring metabolites pre- and post-TIPS. **c** Network representation of hepatic ChemProp scores highlighting high-scoring clusters. **d, e** Specific examples of metabolite pairs within (**d**) glycerophosphocholines (GPC) and (**e**) bilirubins clusters that display high ChemProp scores. Metabolite ID and m/z are shown for metabolites, and associated delta m/z shown for metabolite pairs. **f** ChemProp scores for pairs of neighboring metabolites that diverged the most on their scores based on HE grade. PIV peripheral vein, HV hepatic vein. **g, h** Examples of 2 metabolite pairs (within bilirubin and glycerophosphocholine subclasses) from panel (**f**) with divergent ChemProp scores based on HE grade. Median is shown for specific metabolites at each timepoint. HE grade: 0 = none (green); 1 = mild (blue); 2+ = severe (red). Source data are provided as a Source Data file.

the new capillaries[9,47]. Our data strongly suggest that patients with a higher degree of intrahepatic shunting are more acclimatized to existing portal toxins prior to TIPS placement, and are therefore less likely to develop worsened HE post-TIPS. This is due to their ability to adjust to the slowly accumulating level of enteric toxins over time as hepatic function deteriorates. Conversely, we hypothesize that less intrahepatic shunting pre-TIPS may lead to an overwhelming influx of portal toxins into the systemic circulation post-TIPs, with subsequent development of HE as a result (Fig. 2e). These findings imply that measurement of intrahepatic shunting pre-TIPS vs. post-TIPS could assist in risk-stratifying patients susceptible to developing high-grade HE months after TIPS.

Methods of measuring intrahepatic shunts in patients with cirrhosis have been described, but have not been updated in decades. These involve imaging using radionucleotide-tagged galactose or ethanol injection, angiography similar to that used for TIPS placement, and histologic stains for angiogenic factors after liver biopsy[5,7,48]. Future work must be devoted to the development of readily available, noninvasive clinical measurements of intrahepatic shunting within cirrhotic individuals before and after TIPS, as it may help determine which patients will be susceptible to the development of HE. Additional studies should include direct evaluation of pre-TIPS shunting, and portal vein samples. This would help determine the relationship between metabolomic differences and the degree of pre-existing intrahepatic shunts, which were not part of our original experimental design due to the surprising and unexpected nature of our results. Nonetheless, our study provides insights into the potential role of pre-existing intrahepatic shunts, metabolome changes, and the development of hepatic encephalopathy post-TIPS.

With the knowledge that intrahepatic shunting likely plays a role in post-TIPS HE and that the most significant metabolomic changes occur among bile acids pre- to post-TIPS, we explored the relationship between bile acids and HE. The change in bile acid abundances in the peripheral and hepatic vein is greater in those who developed HE (Fig. 3c). This finding was confirmed upon examining bile acid changes as they relate to HE grade in the peripheral vein, which showed three conjugated hydroxylated bile acids were inversely correlated with the later development of HE (Fig. 4a). Patients who were hospitalized and treated for HE exacerbation had fluctuations in these specific bile acids that were consistent with their disease course (Fig. 4b, c).

Combined, our results suggest that the increased risk of developing HE post-TIPS is related to the extent to which the bile acid abundance pool changes immediately post-TIPS. Closer inspection of these bile acids leads us to suspect that they are secondary conjugated glycine and taurine bile acids, with mtb 181 possibly corresponding to deoxycholic acid (DCA) or one of its isomers. These results suggest that bile acids play a protective role against the development of HE. Participants who did not develop post-TIPS HE did not demonstrate the decline in bile acids (Figs. 3c and 4), as they likely had intrahepatic shunting at the time of TIPS and had already acclimated to the portal toxins within systemic circulation. Those without intrahepatic shunting pre-TIPS are then exposed to the alterations in metabolites post-TIPS and are unable to further compensate for the portosystemic shunting. Of note, the methodology used here provides a snapshot of metabolites possibly implicated in HE. It is possible that toxins other than those captured by our method contribute to cognitive dysfunction in HE.

Previous studies in animal models have shown that bile acids can offer a therapeutic advantage. The neuronal oxidative damage induced by bilirubin can be inhibited by supplementation with the bile acid glycoursodeoxycholic acid (GUDCA)[49–53]. Indeed, the therapeutic, anti-inflammatory effects of GUDCA and TUDCA have been described in many disorders including insulin resistance, glucose intolerance, Barrett's esophagus, retinal disease, and neurodegenerative diseases such as amyotrophic lateral sclerosis[54–59]. GUDCA directly inhibited the farnesoid X receptor, a ligand-mediated nuclear receptor whose activation is linked to HE and whose inhibition decreased HE symptoms in mice[54,60]. Our study links bile acids as potentially protective against HE in humans.

In addition to determining changes pre- to post-TIPS in the context of overall metabolome and specific metabolite features, we also investigated how TIPS can impact pairs of related metabolites and chemical transformations between them. This was possible due to recent advances in the field, and the use of FBMN of untargeted metabolomics with time-series data. The most prominent mass changes in neighboring metabolites of hepatic vein occur among the glycerophosphocholine and bilirubin clusters (Fig. 5d–g). Modifications of glycerophosphocholine, a precursor to the important neurotransmitters acetylcholine and choline, have been previously described in patients with HE[61–63]. However, the link between biliverdin and HE has not explicitly been studied. Biliverdin is reduced to bilirubin via biliverdin reductase A, and there is a strong relationship between HE and free bilirubin[64–71]. Our data show that possible oxidation reactions involving biliverdin occur to a greater degree in participants who developed high-grade HE. Interestingly, free bilirubin is rapidly solubilized by bile acids in vitro, perhaps explaining the potential protective effects of bile acids observed in our study in the context of bilirubin metabolism (Fig. 5g)[72].

At present, HE is managed with the enteric pharmaceuticals, rifaximin and lactulose[73,74]. These drugs affect the composition of the gut microbiome, thus implicating microbial metabolites or products as potential agents that can influence the onset of HE— an observation supported by fecal microbiota transplant studies[1,75]. The relationship between the bile acid pool and gut microbiome is well-established, but the role of the microbiome in HE is not thoroughly understood[76–79]. Future areas of research should explore how the bile acid, glycerophosphocholine, and bilirubin metabolites in our study relate to the microbiome by performing fecal collections before and after the TIPS procedure.

There are a few noteworthy limitations to our study. Our approach has limited coverage of the metabolic space, and not all metabolites are captured by the extraction and LC-MS/MS methods. Furthermore, the annotations of most compounds are based on the spectral libraries, many of which only allow for reliable subclass annotation as opposed to specific compounds (identified by MS/MS matching to reference library)[80]. We are also making an assumption that the toxin inducing HE is a metabolite, but it could be a protein or another signal coming from the gut or inflammatory system. In addition, this study would benefit from being supplemented by association analyses between the microbiome and luminal metabolomics data to verify that our metabolite hits are of microbial origin. Our results should also be further verified in larger cohorts of patients, especially those who are being hospitalized and treated for HE, to determine whether these bile acids are universally decreased in HE and are potential biomarkers for development of post-TIPS HE. Furthermore, since we show that intrahepatic shunting could be playing a protective role against HE, future studies should look into methods to directly measure the extent of intrahepatic shunting before TIPS and evaluate portal vein blood. Bedside-to-bench experiments should further characterize the chemical structure of these bile acids, study their effects on neuroinflammation, and determine whether their supplementation could result in a treatment for HE.

There are also some limitations with the diagnosis of HE grade 1 (covert HE) such as interobserver reliability and the episodic nature of the symptoms, which may introduce more noise into our analysis. However, we mitigate the effect of this by using the worst HE grade recorded. Our results also show that HE grade 1 consistently falls in between the metabolome and metabolite levels of HE grade 0 and 2+. Moreover, removal of HE grade 1 from analysis, either by exclusion or by grouping it with the HE grade 0 (i.e., covert HE group) does not affect the conclusions of the study. These factors show the robustness and consistency of these findings with the recorded HE grades.

In summary, this study employed the innovative methodology of untargeted metabolomics to identify changes in metabolites in participants with cirrhosis undergoing TIPS. Relating these changes to HE highlighted physiological differences between those that developed HE and those that did not, which led us to propose that intrahepatic shunting may be occurring in patients who develop HE post-TIPS. In two patients who were readmitted for HE exacerbations during our study period, we observed decreases in bile acids immediately post-TIPS, implying a protective mechanism for these metabolites. Furthermore, we demonstrated that glycerophosphocholines and bilirubins, vital neuromodulating metabolites, fluctuate within our participants' hepatic circulation post-TIPS. Ultimately, the insights described herein will aid in identifying potential biomarkers for HE, as well as pave the way for therapeutic targets in prevention and treatment of this highly morbid condition.

## Methods
### Cohort and study design
This is a multi-center prospective cohort study approved by the IRBs of the University of California, San Diego, and University of Florida (Supplementary Fig. 1). The study complied with all relevant ethical regulations. Patients scheduled to undergo elective transjugular intrahepatic portosystemic shunt (TIPS) due to severe side effects related to portal hypertension were identified by participating interventional radiology physicians prior to the procedure. Patients undergoing TIPS were screened between August 2018 and February 2019. Inclusion criteria were met if patients had cirrhosis and were undergoing elective TIPS, had an absence of overt HE at the time of enrollment, were aged >18, were not pregnant, and were willing and able to consent to the study. Patients were excluded if found to have non-cirrhotic portal hypertension, other potential causes of cognitive deficits, a previous liver transplant, or were prescribed medications that could cause changes in the bile acid pool (e.g. ursodiol, sequestrants). Once study participants were selected for enrollment, informed consent was obtained by a study coordinator. A total of 22 individuals participated in the study. Population characteristics included: mean age (±SEM) of $59.8 \pm 2.4$; sex/gender ($n = 8$ female; $n = 14$ male). There was no discordance between self-reported sex and gender. Participants did not receive compensation for participation in the

study. On the day of the procedure and pre-anesthesia, peripheral vein blood was drawn from the patients pre-TIPS. During the procedure, the interventional radiologist collected hepatic vein blood just prior to and immediately after shunt placement, described as pre-TIPS and post-TIPS hepatic vein, respectively. Hepatic pressure in mmHg was recorded during TIPS as follows: gradient pre-TIPS was measured as the direct pre-stent portal vein pressure; gradient post-TIPS was measured as the direct post-stent (and post esophageal varix embolization, if any) portal vein pressure. Participants from facility A received one 3.375 g intravenous piperacillin-tazobactam dose intraoperatively while participants from facility B received 2 g intravenous cefazolin. Hepatic vein samples were collected under similar conditions during the TIPS procedure while patients were under anesthesia. After the procedure but prior to admission to the post-anesthesia unit, the participants provided another peripheral vein blood sample (post-peripheral vein). Finally, participants provided a fasting blood sample on the day of discharge with their morning labs (before discharge). Overall, the time difference between pre-peripheral vein to post-peripheral vein ranged from 1 to 3 h and post-peripheral vein to before discharge from 14 to 19 h. The blood samples were centrifuged at $1000 \times g$ for 15 min at room temperature to obtain plasma and buffy coat, which were then aliquoted and stored at −80 °C.

Upon discharge, participants' hepatology assessments were retrospectively reviewed for up to one year after the TIPS procedure. HE grade (as defined by the West Haven Criteria) was determined from both scheduled visits and visits prompted by symptoms. The most severe HE grade documented during these visits was used in the analyses. When not explicitly stated in the visit documentation, the West Haven criteria were employed to calculate HE grade, as this system is well-established in the field of hepatology. Notably, the diagnosis of HE grade 1 is limited by this retrospective chart review, as psychometric analyses were not always performed by clinicians during these visits, and HE grade was assigned based on clinical assessment. However, while psychometric evaluation is useful for clinical diagnosis of HE grade 1, it is not explicitly required nor is it completely reliable, particularly within diverse populations[81,82].

Participants were prospectively monitored for readmission to the hospital for overt signs of HE and, when possible, additional peripheral blood samples were collected during those visits. HE treatment of these participants post-discharge was at the discretion of the participants and their hepatologists, with changes in medication typically occurring in the setting of clinically significant HE. REDCap 13.8.1 - © 2023 Vanderbilt University and Microsoft Excel Version 16.75 were used for data collection.

After the data were collected, the hospital courses of the two participants who were readmitted during the study period were examined. Participant 11_a (Fig. 3c) was a 68-year-old woman with a history of cryptogenic cirrhosis with banded esophageal varices but with no prior history of HE. She underwent a TIPS procedure due to progressively worsening ascites refractory to diuretic therapy and the need for intermittent paracentesis. She was readmitted to the medical intensive care unit 9 days post-TIPS with obtundation requiring intubation for airway protection. Blood was drawn on the day of admission before initiation of hepatic encephalopathy treatments (pre-HET, Fig. 3c). The ammonia level on admission was 140 µmol/L. She was given lactulose and rifaximin for HE treatment, to which she responded well. Blood was collected for analysis on the day after treatment initiation (post-HET, Fig. 3c). She was discharged 9 days later. However, she was readmitted within 72 h for altered mental status. Ammonia level at the time was 169 µmol/L, but she was less encephalopathic than during the prior admission and did not require intubation. Her encephalopathy resolved after continuous polyethylene glycol infusion via a nasojejunal tube, but was again admitted 352 days post-TIPS for HE in the setting of urosepsis.

Participant 12_a (Fig. 3b) was a 62-year-old man with a history of alcoholic cirrhosis and esophageal varices, and with no history of HE. He underwent TIPS placement with simultaneous coil embolization of a left gastric varix. On post-TIPS day 30, he became acutely altered and was admitted to a nearby hospital, where the ammonia level was 123 µmol/L. He was started on lactulose, but due to worsening encephalopathy, was transferred to the tertiary facility participating in the study. On admission to the participating facility, his ammonia level was 9 µmol/L, despite ongoing encephalopathy. Blood was collected before the initiation of hepatic encephalopathy treatments (pre-HET, Fig. 3b). He underwent a nasojejunal tube placement and was given polyethylene glycol continuously, in addition to lactulose and rifaximin until his altered mentation resolved. Blood was again collected for analysis (post-HET, Fig. 3b).

Besides the two participants who were hospitalized, no additional blood samples were collected post discharge.

## Untargeted metabolomics by LC-MS/MS
For untargeted LC-MS/MS, 400 µl of pre-chilled extraction solvent (100% MeOH with 1.25 µM sulfamethazine) was added to 100 µl of plasma. Samples were briefly vortexed and then incubated at −20 °C for 20 min for methanol extraction, after which they were centrifuged for 15 min at $20,000 \times g$. The supernatant was transferred to a pre-chilled 96-well deep well plate. Samples were dried using a centrifugal low-pressure system for 8 h. Dried extracts were stored in sealed plates at −80 °C until analysis. Samples were resuspended in methanol (MeOH) with 1.5 µM of sulfadimethoxine, vortexed, sonicated, transferred to a shallow 96-well plate, and diluted 2X. Untargeted metabolomics analysis was conducted using an ultra-high-performance liquid chromatography system (UltiMate 3000; Thermo Fisher Scientific, Waltham, MA) coupled to a Maxis quadrupole time of flight (Q-TOF) mass spectrometer (Bruker Daltonics, Bremen, Germany) with a Kinetex C18 column (Phenomenex, Torrance, CA, USA). Data were collected as described in Gauglitz et al. 2020 in positive electrospray ionization mode[83]. All solvents used were liquid chromatography-mass spectrometry [LC-MS]-grade (Fisher Scientific). Raw data were exported to open format. *mzXML* files using DataAnalysis (Bruker) and files were uploaded to the GNPS platform.

## Metabolomics data processing and analysis
Feature based molecular networking (FBMN) version release_28.2 was performed on the GNPS platform using pre-processed MZmine2 files from LC-MS/MS experiments and is available under the following link: https://gnps.ucsd.edu/ProteoSAFe/status.jsp?task=8068b7cb58f2465f97a15b3ce30d593c. In brief, MS/MS fragment ions ±17 Da of the precursor m/z were removed and the top 6 fragment ions within a ±50 Da window were selected. Dataset and analysis files, including MZmine2 parameters, are available on MassIVE (dataset MSV000090443). Edges of the molecular network were created based on a filter of 0.65 cosine score minimum and more than 5 matched peaks. GNPS spectral libraries were used to match spectra in the network to get level 2 and level 3 annotations[80]. Cytoscape 3.8.2 was used to visualize networks. In addition, a chemical proportionality (ChemProp version release_25) approach was applied to the longitudinal data to calculate changes in abundance between neighboring nodes in order to identify changes in the abundances between every two neighboring nodes. The ChemProp score is calculated through the $\log_{10}$-transformed ratio of peak area proportions of two neighboring molecular network nodes between two sequential times points[39]. This score reflects the level of the change in abundance pre- to post-TIPS for a given pair of neighboring metabolites, where a high score indicates greater changes, and the sign indicates directionality. Furthermore, inspecting the mass change associated with each pair and its ChemProp strength can indicate potential biological or chemical

transformations within the network, which can be visualized on a dataset scale to prioritize modification patterns[84].

## Metabolomics statistical analyses

To calculate metabolomic dissimilarities across participants, we used the robust aitchison β-diversity calculated using DEICODE in Qiime2 v 2021.4 (--p-min-feature-count 10 and --p-min-sample-count 500) and plotted against robust principal component (RPCA) plots[85,86]. We used a PERMANOVA test to find if there were significant differences between the metabolomes of subjects based on shunt placement. There was no significant metabolome difference based on sex/gender (peripheral vein $P = 0.348$; hepatic vein $P = 0.694$). To find what metabolites were driving the dissimilarity between post-peripheral vein and before discharge in the RPCA, we used the tool Qurro in Qiime2 v 2021.2 to find what metabolite groups were most contributing to the directionality in the first principal component (PC1)[41]. Once we identified bile acids as being clustered to one side of PC1, we used Qurro to calculate the log ratio of bilirubins (a metabolite spread across PC1) to bile acids per sample. For each metabolite, we ran a paired Wilcoxon test on the log10(abundance) at pre-peripheral vein vs. post-peripheral vein, post-peripheral vein vs. before discharge, and pre-peripheral vein vs. before discharge, or pre-hepatic vein vs. post-hepatic vein using a FDR correction of 0.1.

We used published data on total BA levels to guide our power analysis. Power analysis during study design was performed based on the assumption that BA levels increase ten-fold in patients with cirrhosis compared to healthy controls (46.0 μM). To determine the number of participants needed, we assumed that the average patient with cirrhosis has a mean total BA level that is tenfold higher than normal (46.0 μM). To be able to detect a difference of 30% change in total BA pool, assuming a within-group standard deviation of 11.0% (based on standard deviation of total BAs in populations of cirrhotic patients) with an $\alpha = 0.05$ and power $(1 - \beta) = 0.8$ (power analysis for paired t-test analysis), we would need a minimum of 5 participants. Since we wanted to have samples from at least 5 participants while they are admitted to the hospital for overt HE, we recruited more participants into the study. The likelihood of a patient post-TIPS being admitted for overt HE is 35%, though it can vary by facility. If we estimate that admission for HE after TIPS at our facility is 25%, we will need to recruit 20 participants to be able to attain 5 samples from patients hospitalized for overt HE. We also performed a retrospective power analysis that showed our sample sizes are sufficient to find a difference of 82% between HE grade 0 and the other groups $(1 - \beta = 0.8; \alpha = 0.05)$. The fact that our actual differences between HE grade 0 and 1 are 391% and HE grade 0 and 2+ are 492% demonstrates that we have more than enough power in our sample size to detect differences in our groups.

To determine statistically significant differences in the levels of bile acids based on the worst HE grade post-TIPS (0, 1, or 2+), we ran a Kruskal–Wallis test on the log10(abundance) of post-peripheral vein metabolites. Change from baseline or change pre/post was calculated as the log10(abundance$_{post}$/abundance$_{pre}$) for each participant's metabolite. Statistical significance was determined for: (a) all changes pre/post based on HE grade using a Kolmogorov–Smirnov test (0 vs. 2+; 0 vs. 1; 2+ vs. 1); or (b) changes pre/post for metabolites within each cluster using a Kruskal-Wallis test (0.2 FDR cutoff) followed by a Wilcoxon test. We used different FDR cut-offs to account for the change in our sample size when subsetting/grouping the data. For box-plots, center lines represent the median, lower and upper limits represent the first and third quartile, and whiskers represent 1.5x of the interquartile range.

Custom scripts were used to plot within-participant metabolome dissimilarities pre- to post-shunt stratified by HE grade (code repository page will be made available for publication). Analyses were performed in Qiime2 version 2021.4 and R version 4.1.0. ChatGPT was used to improve the clarity and accessibility of portions of the manuscript.

## Reporting summary

Further information on research design is available in the Nature Portfolio Reporting Summary linked to this article.

## Data availability

The raw and processed untargeted metabolomics data generated in this study have been deposited in the MassIVE database under the MassIVE ID MSV000090443 [https://massive.ucsd.edu/ProteoSAFe/dataset.jsp?task=672015362dca49278d517bba9a4a00e0]. Source data are provided with this paper.

## Code availability

Code to perform analysis of metabolomics data is available on GitHub [https://github.com/acdm2020/HE-TIPS/releases/tag/v1.0] and Zenodo [DOI: 10.5281/zenodo.8206272].

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

## Acknowledgements

Initial funds for the studies were provided by a pilot and feasibility grant from UL1 TR001442. A.C.D.M. is supported by R01 HL148801-02S1. S.F.R. was supported by the PD Soros Foundation. D.P. was supported through the German Research Foundation (DFG) through the CMFI Cluster of Excellence (EXC 2124) and the Collaborative Research Center CellMap (TRR 261). Al.Z. is supported by NIH K08 DK113244 and the UF Foundation-Robert Duggan Fund. Am.Z. is supported by Kavli Institute for Brain and Mind at UC San Diego, AASLD Liver Scholar Award, VA Merit BLR&D Award I01 BX005707, and NIH K08 DK102902, R03 DK114536, R21 MH117780, R01 HL148801, R01 EB030134, and U01 CA265719. All UC San Diego authors receive institutional support from NIH P30 DK120515, P30 DK063491, P30 CA014195, P50 AA011999, and UL1 TR001442. The funders had no role in study design, data collection and interpretation, or the decision to submit the work for publication. The contents do not represent the views of the U.S. Department of Veterans Affairs or the United States Government.

## Author contributions

Conceptualization - Am.Z.; Methodology - A.C.D.M., J.M.G., A.A., D.P., Al.Z., Am.Z.; Data/Sample acquisition - Al.Z., U.B.K., M.L., H.A., A.B.G., C.W.; Formal analysis - S.F.R., A.C.D.M.; Investigation - A.C.D.M., S.F.R., J.M.G.; Resources - Al.Z., Am.Z., P.C.D.; Data curation - A.C.D.M., S.F.R., J.M.G., A.A., D.P., P.C.D.; Writing (Original Draft) - S.F.R., A.C.D.M., A.M.F.; Writing (Review & Editing) - J.M.G., D.P., A.A., Al.Z., Am.Z., U.B.K., M.L., A.B.G., H.A., I.V., C.W., P.C.D.; Visualization - A.C.D.M., S.F.R.; Supervision - Al.Z., Am.Z.; Project administration - Al.Z., Am.Z.; Funding acquisition - Al.Z., Am.Z.

## Competing interests

Am.Z. is a co-founder, acting chief medical officer, and equity-holder in Endure Biotherapeutics. P.C.D. is a scientific advisor and holds equity to Cybele and a Co-founder, advisor and holds equity in Ometa, Arome and Enveda with prior approval by UC-San Diego and consulted in 2023 for DSM animal health. A.A.A. is a founder of Arome Science Inc. The remaining authors declare no competing interests.

## Additional information

[1]Division of Gastroenterology and Hepatology, University of California, San Diego, La Jolla, CA, USA. [2]Biomedical Sciences Graduate Program, University of California, San Diego, La Jolla, CA, USA. [3]Department of Pediatrics, University of California, San Diego, La Jolla, CA, USA. [4]Skaggs School of Pharmacy and Pharmaceutical Sciences, University of California, San Diego, La Jolla, CA, USA. [5]Department of Surgery, College of Medicine, University of Florida, Gainesville, FL, USA. [6]CMFI Cluster of Excellence, Interfaculty Institute of Microbiology and Infection Medicine, University of Tübingen, Tübingen, Germany. [7]Department of Chemistry, University of Connecticut, Storrs, CT, USA. [8]Department of Radiology, Division of Interventional Radiology, College of Medicine, University of Florida, Gainesville, FL, USA. [9]Division of Gastroenterology, Department of Medicine, Veterans Affairs Puget Sounds Health Care System, Seattle, WA, USA. [10]Division of Gastroenterology, Department of Medicine, University of Washington, Seattle, WA, USA. [11]San Diego Imaging, San Diego, CA, USA. [12]Departments of Radiology, University of California San Diego Medical Center, La Jolla, CA, USA. [13]Center for Microbiome Innovation, University of California, San Diego, La Jolla, CA, USA. [14]Department of Computer Science and Engineering, University of California, San Diego, La Jolla, CA, USA. [15]Center for Computational Mass Spectrometry, University of California, San Diego, La Jolla, CA, USA. [16]Department of Biochemistry and Molecular Biology, College of Medicine, University of Florida, Gainesville, FL, USA. [17]J. Crayton Pruitt Family Department of Biomedical Engineering, Herbert Wertheim College of Engineering, University of Florida, Gainesville, FL, USA. [18]Jennifer Moreno Department of Veterans Affairs Medical Center, La Jolla, CA, USA. [19]Institute of Diabetes and Metabolic Health, University of California, San Diego, La Jolla, CA, USA. [20]These authors contributed equally: Ana Carolina Dantas Machado, Stephany Flores Ramos. ✉e-mail: ali.zarrinpar@surgery.ufl.edu; azarrinpar@ucsd.edu

