## [Peer Review File · Nature Communications]

Portosystemic Shunt Placement Reveals Blood Signatures for the Development of Hepatic Encephalopathy through Mass SpectrometryReviewers' comments:

Reviewer #1 (Remarks to the Author):

The paper reports the results of an untargeted metabolomic analysis performed on blood sampled from hepatic and peripheral veins in patients undergoing TIPS. The aims were: 1) to understand which metabolite levels change because of portosystemic shunt placement and 2) to determine whether these changes can predict HE.

Both aims are extremely relevant because the pathogenesis of hepatic encephalopathy is incompletely known and because the strong pathophysiological considerations suggest that compounds coming from the intestine and not metabolized by the liver because of parenchymal damage and portosystemic shunts may be responsible for the occurrence of hepatic encephalopathy.

Thus, the approach of studying patients submitted to a TIPS is interesting, because during the procedure the blood directly drawing from the hepatic vein can be collected and analyzed.

The most surprising result of the study is the small difference observed between the metabolome collected in the hepatic vein before and after the opening of the shunt. The authors suggest that this result may depend on the presence of significant intrahepatic shunt already present before the TIPS, which is certainly reasonable.

As the direct evaluation of the degree of the pre-TIPS shunting is lacking, the authors hypothesized that the patients who already have significant intrahepatic shunting have a low dissimilarity in the metabolome collected in the hepatic vein before and after TIPS. In my opinion the possibility of establishing the amount of portal blood shunted before TIPS on the dissimilarity of the metabolome remains tentative taking into consideration the methodology used. During the TIPS procedure it is possible to directly collect the portal blood before the opening of the shunt and I do not understand why these samples were not taken and compared to the metabolome of the hepatic vein to estimate the amount of blood shunted. The comparison between portal and hepatic blood could have provided better information on the presence and the quantitative relevance of the shunts already present before TIPS.

Based on the above proposed interpretation, the authors hypothesized that patients who

develop more severe HE have less intrahepatic shunting before the TIPS procedure. To test this hypothesis, they correlated the metabolome dissimilarity to the degree of hepatic encephalopathy developed during the follow up.

Here there are, in my opinion, the most relevant methodological problems. In fact, in the Materials and Methods section how the degree of HE was evaluated is insufficiently described and probably inappropriate. Some questions are: when were the patients evaluated for the detection of HE during the follow up? It was performed during a scheduled visit or because the patients had symptoms? Were they submitted to any treatment after discharge? There was any precipitating event at the time of the onset of HE. In any case, HE was graded as Grade 1 in 10 patients. This led to an unusually high incidence of HE after TIPS (77%) but the most important point is that Grade 1 HE is difficult to be detected and for this reason it is currently included into the so called "covert HE" a diagnosis not possible without any psychometric evaluation. Even in the reference cited for support the method used to score HE, grade 1 is considered based on clinical finding usually not reproducible. As most of the results and their explanation are based on the correlation between metabolomic data and the degree of HE, the uncertainty in HE evaluation is relevant for the result interpretation and consequently most of the conclusions proposed are arguable.

An example of these methodological problem is depicted in figure 2, panel B, which reports the portal pressure changes pre- to post-TIPS by HE grade. No relationship was found. This is an unusual result which may depend on the above illustrated uncertainty in the HE grading as well as on the measure of the portal and hepatic pressure obtain to establish the portosystemic pressure gradient. In fact, two different methodologies were used to obtain the gradient before the shunt opening (by HVPG, which an indirect estimation of the portal pressure) and after the shunt opening (by direct pressure in the portal vein and in the hepatic vein). Usually, the two pressures are measured directly in the portal and in the hepatic vein before and after the shunt opening. Again, I do not understand why the portal vein was not used before the TIPS either to collect the blood and to measure the portal pressure.

Thus, the hypothesis that patients who develop more severe HE have less intrahepatic shunting before the TIPS procedure is, in my opinion, based on results obtained with

arguable methodology.

The conclusion proposed in the Discussion, i.e. “that patients with a higher degree of intrahepatic shunting are more acclimatized to the portal toxins prior to TIPS placement” and are therefore less likely to develop HE post-TIPS is arguable and somewhat naïve. The literature has definitely established a relationship between the presence of portosystemic blood shunting and HE (HE for a long time was named “portosystemic encephalopathy”). Here only the presence intrahepatic portosystemic shunt (supposed to be present but not directly measured) is taken into considerations. Portal hypertension produces several portosystemic shunts, they may be intrahepatic but are also certainly extrahepatic and leading to the exposure of portal toxins. How can a patient with a high degree of intrahepatic of portosystemic shunt be acclimatized to the portal toxins when the portal toxins mainly derive from the extrahepatic portosystemic shunt, remains uncertain.

Unfortunately, the methodological problems illustrated above question the relevance of bile acids which may be involved in the pathogenesis of HE.

I would like to encourage the authors to solve the methodological problems of the paper because their approach to the pathogenesis of HE is undoubtedly very interesting.

Minor points

Reading of the paper is very difficult and the abbreviations used increase the difficulty. For example, the peripheral vein is abbreviated as PIV, why?

Reviewer #2 (Remarks to the Author):

This interesting paper considers metabolite changes, measured by mass spectrometry techniques in hepatic venous blood and peripheral venous blood, as a result of transjugular intrahepatic portosystemic shunt (TIPS). Mass spectrometry analyses suggested decreases in bile acids in peripheral venous blood post TIPS vs pre TIPS. Further, those patients who subsequently developed a more severe hepatic encephalopathy (HE) after TIPS showed a greater decrease in bile acids and glycerophosphocholine levels post TIPS vs pre TIPS in peripheral venous blood samples. A suggested interpretation of these findings was that intrahepatic shunting may be occurring in patients who develop HE post-TIPS. Conversely, those patients with established intrahepatic shunting pre-TIPS were less likely to develop HE post-TIPS.

Mass spectrometry analyses of hepatic venous blood and peripheral venous blood pre- and post-TIPS provide a novel snapshot of metabolite levels at two specific time points. As the authors comment, their assumption that mass spectrometry measurable metabolites are the toxins responsible for inducing HE post-TIPS may not be valid.

The authors have interpreted their observation of surprisingly small differences between the pre- and post-hepatic vein mass spectrometry metabolome as suggesting considerable intrahepatic shunting was already occurring in most participants prior to undergoing TIPS. Have the authors considered the impact of surgery as a confounding factor on the hepatic vein metabolome, which may be masking changes due to TIPS? Appreciating that efforts to characterise intrahepatic shunting have previously been sparse, did the authors consider an independent measurement of this phenomenon in their cohort pre-TIPS? Additional evidence of the presence or absence of intrahepatic shunting pre-TIPS would strengthen the interpretation that intrahepatic shunting may be occurring in patients who develop HE post-TIPS.

The Abstract focuses on the background to the study rather than describing the patient cohort, the techniques used or discussing the hypothesis of less hepatic shunting pre-TIPS as an indicator development of HE post-TIPS.

The patient cohort (Table S1) comprised of eight female and 14 male subjects. Were any sex/gender based sub-analyses of the mass spectrometry findings undertaken? As this was a multi-centre study, were any differences noted in the mass spectrometry findings pre-TIPS between the two centres?

Minor points: The terminology of pre-/post-HV, pre-/post PIV and BDC would benefit from being clearer. Some of the references are incomplete. What was the time between blood sampling and sample centrifugation? Table S4 should be mentioned in the text.

RESPONSE TO REVIEWERS

Reviewer #1 (Remarks to the Author):

The paper reports the results of an untargeted metabolomic analysis performed on blood sampled from hepatic and peripheral veins in patients undergoing TIPS. The aims were: 1) to understand which metabolite levels change because of portosystemic shunt placement and 2) to determine whether these changes can predict HE.

Both aims are extremely relevant because the pathogenesis of hepatic encephalopathy is incompletely known and because the strong pathophysiological considerations suggest that compounds coming from the intestine and not metabolized by the liver because of parenchymal damage and portosystemic shunts may be responsible for the occurrence of hepatic encephalopathy.

Thus, the approach of studying patients submitted to a TIPS is interesting, because during the procedure the blood directly drawing from the hepatic vein can be collected and analyzed.

We would like to thank the reviewer for highlighting the relevance of our work in the study of hepatic encephalopathy pathogenesis. We appreciate the recognition of the relevance of our aims, which focus on understanding the pathogenesis of hepatic encephalopathy and the potential predictive value of changes in metabolite levels following TIPS placement. We agree that the unique approach of studying patients undergoing TIPS allows for direct analysis of hepatic vein blood, and we are grateful for the appreciation of this aspect of our work.

1. The most surprising result of the study is the small difference observed between the metabolome collected in the hepatic vein before and after the opening of the shunt. The authors suggest that this result may depend on the presence of significant intrahepatic shunt already present before the TIPS, which is certainly reasonable. As the direct evaluation of the degree of the pre-TIPS shunting is lacking, the authors hypothesized that the patients who already have significant intrahepatic shunting have a low dissimilarity in the metabolome collected in the hepatic vein before and after TIPS. In my opinion the possibility of establishing the amount of portal blood shunted before TIPS on the dissimilarity of the metabolome remains tentative taking into consideration the methodology used. During the TIPS procedure it is possible to directly collect the portal blood before the opening of the shunt and I do not understand why these samples were not taken and compared to the metabolome of the hepatic vein to estimate the amount of blood shunted. The comparison between portal and hepatic blood could have provided better information on the presence and the quantitative relevance of the shunts already present before TIPS

We thank the reviewer for bringing up this very important point. The reviewer points out that the most surprising result from this study is the small differences in the metabolome of hepatic vein blood pre- and post-TIPS. As this unexpected result was not part of our original hypothesis, direct evaluation of pre-TIPS shunting was not included in our study design nor included in our power analysis or IRB submission. However, the differences

in metabolome dissimilarity between patients with different HE grades suggest that the metabolome of patients who develop HE within a year after shunt placement is already inherently more dissimilar than that of those who do not develop HE. These results suggest that intrahepatic portosystemic venous shunts were present before TIPS and contributed to portal blood shunting. We elaborate on each of these points below:

- a. As the reviewer correctly identified, **our primary study outcome** was to determine differences in metabolites before and after TIPS placement. If we had done the design of the experiment in the way that the reviewer suggested (collection of portal blood), we would not be able to address the main goals of our study, which the reviewer also seems to value. Furthermore, we don't have any evidence (both experimental or in the literature) to suggest that post-TIPS right hepatic vein blood is appreciably different from right portal vein blood. Nonetheless, the main results and conclusions we derived from the hepatic vein blood – the lack of metabolomic differences before and after shunt placement and relation to HE severity – are relevant and independent of results we would have acquired directly from the portal vein prior to TIPS.
- b. The reviewer raises a valid point that having portal vein blood would have allowed us to directly compare the metabolomes of the portal and hepatic veins. However, as pointed out by the reviewer, our findings in the hepatic vein were surprising and unexpected, and thus collection of portal vein blood prior to shunt was not part of our study design and beyond the scope of our original study. As such, unexpected findings cannot be anticipated or designed for. In this case, this is the result of the experimental process and indicates a novel phenomenon deserving additional inquiry. We agree with the reviewer that this is an important finding and one that should be used to design and justify additional studies and hence should be published.

To address this comment, we have made the following modifications to the discussion to clarify our goals and the nature of intrahepatic shunting:

In the Discussion, we have included a paragraph acknowledging the limitations of our study, specifically highlighting the lack of direct evaluation of pre-TIPS shunting and the absence of portal vein blood samples. We also emphasize that our results were unexpected and that the study was not designed to investigate intrahepatic shunting.

Additionally, we have added a section in the Discussion suggesting future studies that could be designed to directly evaluate intrahepatic shunting and the amount of portal blood shunted before TIPS. We propose that these studies should include collection of portal vein blood and measures of intrahepatic shunting before shunt placement, as well as investigate the relationship between metabolomic differences and the degree of pre-existing intrahepatic shunts.

Finally, we have emphasized that despite the limitations and unexpected results, our study still provides valuable insights into the relationship between the metabolome and the development of hepatic encephalopathy after TIPS placement, as well as the potential role of pre-existing intrahepatic shunts.

We hope that these changes address the reviewer's concerns and clarify our study's goals, findings, and limitations. We agree that further studies are needed to better understand the role of intrahepatic shunting in the context of TIPS placement and its impact on the metabolome.

2. Based on the above proposed interpretation, the authors hypothesized that patients who develop more severe HE have less intrahepatic shunting before the TIPS procedure. To test this hypothesis, they correlated the metabolome dissimilarity to the degree of hepatic encephalopathy developed during the follow up. Here there are, in my opinion, the most relevant methodological problems. In fact, in the Materials and Methods section how the degree of HE was evaluated is insufficiently described and probably inappropriate. Some questions are: when were the patients evaluated for the detection of HE during the follow up? It was performed during a scheduled visit or because the patients had symptoms? Were they submitted to any treatment after discharge? There was any precipitating event at the time of the onset of HE. In any case, HE was graded as Grade 1 in 10 patients. This led to an unusually high incidence of HE after TIPS (77%) but the most important point is that Grade 1 HE is difficult to be detected and for this reason it is currently included into the so called "covert HE" a diagnosis not possible without any psychometric evaluation. Even in the reference cited for support the method used to score HE, grade 1 is considered based on clinical finding usually not reproducible. As most of the results and their explanation are based on the correlation between metabolomic data and the degree of HE, the uncertainty in HE evaluation is relevant for the result interpretation and consequently most of the conclusions proposed are arguable.

We apologize for the lack of clarity in the methodology. We have now revised the Materials and Methods section to provide more details on the evaluation of HE during the follow-up period and the treatment of patients after discharge:

- a. *"In fact, in the Materials and Methods section how the degree of HE was evaluated is insufficiently described and probably inappropriate. Some questions are: when were the patients evaluated for the detection of HE during the follow up? It was performed during a scheduled visit or because the patients had symptoms?"*

We have added a passage to the Methods explaining that we reviewed the participants' hepatology assessments for up to one year after the TIPS procedure. We have clarified that HE grades were determined from both scheduled visits and visits prompted by symptoms. We have also mentioned that the worst HE grade during this follow-up period was used in the analysis for this

study, which may not have been clear. In addition, we prospectively monitored whether the participants were admitted to the hospital for HE. When possible, additional samples (e.g. main manuscript **Fig 1A**) were collected from these hospital stays.

- b. *“Were they submitted to any treatment after discharge? There was any precipitating event at the time of the onset of HE.”*

We have included information on the treatment of patients after discharge in the Methods. We explain that the addition of treatment for HE was at the discretion of the participants and their hepatologists, with changes in medication typically occurring in the setting of clinically significant HE. Precipitating events that may have affected the addition of medications are difficult to assess retrospectively. However, the two patients who were admitted for HE had increased portosystemic gradients which were treated with TIPS revisions. It is important to note that, except for the patients who were hospitalized for HE, no additional blood was collected from the individuals in the study and that the metabolomics results are solely from the time of the TIPS procedure.

- c. *“In any case, HE was graded as Grade 1 in 10 patients. This led to an unusually high incidence of HE after TIPS (77%) but the most important point is that Grade 1 HE is difficult to be detected and for this reason it is currently included into the so called “covert HE” a diagnosis not possible without any psychometric evaluation. Even in the reference cited for support the method used to score HE, grade 1 is considered based on clinical finding usually not reproducible. As most of the results and their explanation are based on the correlation between metabolomic data and the degree of HE, the uncertainty in HE evaluation is relevant for the result interpretation and consequently most of the conclusions proposed are arguable.”*

We appreciate the reviewer's concern about the accuracy and reproducibility of HE Grade 1 diagnosis, which is critical for the interpretation of our results. In light of these concerns, we have made the following changes to our manuscript:

1. In the Methods section, we have clarified that the West Haven criteria were used to diagnose patients with HE Grade 1 (PMID: 25042402). We have also noted that while psychometric evaluation is useful for minimal HE, it is not required or necessary for the clinical diagnosis of grade 1 (covert) HE.
2. In the Discussion, we have included a paragraph addressing the limitations associated with the diagnosis of HE Grade 1, such as interobserver reliability and the episodic nature of the symptoms. We have explained that our study focused on the patients' worst HE grade after TIPS placement, which helps to mitigate the potential impact of these limitations on our results and conclusions.
3. We have also emphasized in the Discussion that the gradation of metabolomic findings in our study consistently shows grade 1 HE results lying between those of grade 0 and grade 2+ samples, which further supports the robustness and consistency of our findings.

4. We have added a statement acknowledging that although the inclusion of the grade 1 cohort may introduce some noise in our statistical analysis, it allows us to demonstrate that our significant findings are robust and reliable in the face of this noise.
5. Lastly, for the benefit of the reviewer, we have added Reviewer Figure 1, which demonstrates that, even when we exclude patients with HE grade 1, or recategorize HE grade 1 to HE grade 0, the key findings of our paper are unchanged.
 - a. For example, the current analysis shows that the difference in the hepatic vein metabolome pre- and post-TIPS remains statistically significant if we only compare metabolomes of participants with HE grades 0 and 2+ ($p=0.024$, see **Rev. Fig.1A** and new **Fig. S4B**). Furthermore, the post-TIPS levels of specific bile acids remain statistically significant if we only consider participants with HE grade 0 and 2+ (**Rev. Fig.1B** and new **Fig. S5A**). Although including HE grade 1 would be more statistically rigorous, we wanted to demonstrate that our results still hold true even when participants with HE grade 1 are excluded.
 - b. In a second analysis, we combined HE grade 1 and HE grade 0 to a single group (HE grade 0/1) to demonstrate a covert vs. overt HE analysis. As demonstrated the pre- and post-TIPS difference between the grades still holds. When we compared the hepatic vein metabolome pre- to post-TIPS for grades HE 0/1 (covert) and 2+ (overt), the results bordered on significance (**Rev. Fig. 1C** and new **Fig. S4C**, $p=0.083$). Additionally, post-TIPS levels of specific bile acids remain statistically significant (**Rev. Fig. 1D** and new **Fig. S5B**).

We hope that these changes address the reviewer's concerns and provide a more balanced discussion of the limitations and strengths of our study. Despite the inherent challenges associated with diagnosing HE Grade 1, we believe our results and conclusions remain valid and informative. Importantly, the additional analyses described in the reviewer figure have been added as supplementary figures **S4B-C** and **S5A-B**. We hope these changes address the reviewer's concerns and contribute to a more robust and balanced discussion of our study."

Reviewer Figure 1. Comparison of metabolome dissimilarity and metabolite levels between HE grades 0, 1 and 2+. A) Pre vs Post TIPS comparison of metabolome dissimilarity between participants with HE grades 0 (n=3) and 2+ (n=6) for the hepatic vein. B) Bile acid levels and significant abundance differences between HE grades 0 (n=4) and 2+ (n=7) in the post-TIPS peripheral blood based on HE grade. C) Pre vs Post TIPS comparison of metabolome dissimilarity between participants with HE grades 0/1 (n=12) and 2+ (n=6) for the hepatic vein. D) Bile acid levels and significant abundance differences between HE grades 0/1 (n=13) and 2+ (n=7) in the post-TIPS peripheral blood based on HE grade. Significance determined based on a Wilcoxon test. Now updated supplement figures 4 (panels A, C) and 5 (panels B, D).

- An example of these methodological problem is depicted in figure 2, panel B, which reports the portal pressure changes pre- to post-TIPS by HE grade. No relationship was found. This is an unusual result which may depend on the above illustrated uncertainty in the HE grading as well as on the measure of the portal and hepatic pressure obtain to establish the portosystemic pressure gradient. In fact, two different methodologies were used to obtain the gradient before the shunt opening (by HVP, which an indirect estimation of the portal pressure) and after the shunt opening (by direct pressure in the portal vein and in the hepatic vein). Usually, the two pressures are measured directly in the portal and in the hepatic vein before and after the shunt opening. Again, I do not understand why the portal vein was not used before the TIPS either to collect the blood and to measure the portal pressure. Thus, the hypothesis that patients who develop more severe HE have less intrahepatic

shunting before the TIPS procedure is, in my opinion, based on results obtained with arguable methodology.

We appreciate the reviewer's concerns regarding our methodology. In light of these concerns, we have made the following revisions to our manuscript:

1. In the Methods section, we have clarified the methodology used for calculating pressure changes before and after shunt placement. We have now recalculated portal pressure changes based on direct pressures measured at the portal vein obtained before and after shunt opening, as recommended by the reviewer.
2. In the Results section, we have updated the text to reflect the new analysis and to indicate that the portal pressure changes between patients with different HE grading remain statistically insignificant with the updated values, consistent with our previous results. We have replaced **Figure 2B** with a new figure (see **Reviewer Figure 2** below), based on the new analysis, that displays the portal pressure changes pre- to post-TIPS by HE grade.

Moreover, an extensive review of the literature did not reveal any previous publication that demonstrates pre- to post-TIPS portal pressure changes at the time of the TIPS as being predictive of HE.

We hope that these changes address the reviewer's concerns and provide a more accurate representation of our methodology and results. We believe that our study still offers valuable insights into the relationship between portal pressure changes and the development of hepatic encephalopathy after TIPS placement, and that our conclusions remain informative despite the inherent limitations of our methods.

Reviewer Figure 2. Portal pressure changes pre- to post-TIPS by HE grade.

Pressure change was calculated as the difference between direct portal vein pressure measurements pre- and post-TIPS. Now updated Figure 2B.

4. The conclusion proposed in the Discussion, i.e. “that patients with a higher degree of intrahepatic shunting are more acclimatized to the portal toxins prior to TIPS placement” and are therefore less likely to develop HE post-TIPS is arguable and somewhat naïve. The literature has definitely established a relationship between the presence of portosystemic blood shunting and HE (HE for a long time was named “portosystemic encephalopathy”). Here only the presence intrahepatic portosystemic shunt (supposed to be present but not directly measured) is taken into considerations. Portal hypertension produces several portosystemic shunts, they may be intrahepatic but are also certainly extrahepatic and leading to the exposure of portal toxins. How can a patient with a high degree of intrahepatic of portosystemic shunt be acclimatized to the portal toxins when the portal toxins mainly derive from the extrahepatic portosystemic shunt, remains uncertain.

We thank the reviewer for bringing up this very important point. We are not aware of any evidence describing that portal toxins are predominantly derived from extrahepatic portosystemic shunts. Our proposition is that, in the presence of intrahepatic shunting, toxin metabolism may be reduced but still present, leading to a slowly accumulating concentration of enteric toxins in the systemic circulation that allows the patient to acclimatize to the level of toxins over time, reducing HE. The role of intrahepatic shunting in toxin metabolism is nicely described in a recent editorial published in *Journal of Hepatology* (PMID: 37178732 emphasis added):

“Clearance of gut-derived ammonia is also hampered by perfusion mismatch between portal blood flow and the hepatic acinar structures devoted to ammonia disposal. Ammonia disposal requires precise compartmentalisation based on the liver architecture: glutaminase activity and urea synthesis occurs in periportal areas, whereas glutamine synthesis in the perivenous hepatocytes. *Structural alterations in the liver architecture in cirrhosis create blood flow/hepatocyte mismatch, as well as intrahepatic shunts that reduce urea synthesis.*”

Indeed, any portosystemic shunting likely contributes to HE. Notably, our study excluded patients who exhibited overt HE at the time of TIPS placement, regardless of whether their encephalopathy was due to toxins derived from intrahepatic or extrahepatic shunts. These patients are ineligible for elective TIPS and were therefore not included in our study (See **Fig. 2E**).

To address the reviewer’s comments, we have made the following change to the manuscript:

1. In the Introduction, we added a passage to describe that portal toxins could be of extrahepatic or intrahepatic origin.

2. In the Discussion, the paragraphs pertaining to intrahepatic shunting were adjusted to first discuss the pathogenesis of intrahepatic shunting, the reduction in metabolism that results, and the subsequent acclimatization to toxins that leads to a lower degree of pre-TIPS HE in intrahepatic shunting.
5. Unfortunately, the methodological problems illustrated above question the relevance of bile acids which may be involved in the pathogenesis of HE.

We have addressed the methodological concerns mentioned above under reviewer comment #2. As a result, we have no reason to believe the methodological approach related to how HE grading was defined would compromise our findings. As a response to the reviewer, we also analyzed bile acid levels between participants with minimal HE (grade 0 or grade 0/1) and those with overt HE (grade 2+) (**Rev. Fig. 1b**). These results indicate decreased levels of bile acids in participants with grade 2+ compared to those with grade 0, independent of participants with grade 1. Thus, these results are consistent with the original findings, which included all participants and showed a decrease in levels of certain bile acids with increased HE severity.

6. I would like to encourage the authors to solve the methodological problems of the paper because their approach to the pathogenesis of HE is undoubtedly very interesting.

We appreciate the reviewer's encouragement and support for our approach to studying the pathogenesis of HE. In response to the feedback provided, we have made the following changes to our manuscript:

1. We have addressed all methodological concerns raised by the reviewer, performed additional analyses, and presented new results that support our original conclusions.
2. In the Discussion, we have further emphasized the need for future studies to investigate the presence of intrahepatic shunting and its role in HE. We have also highlighted the importance of addressing the methodological issues identified by the reviewer in order to advance our understanding of the complex mechanisms involved in the development of HE after TIPS placement.

We are confident that these revisions have substantially improved the quality and clarity of our manuscript. We believe that our study provides valuable insights into the pathogenesis of HE and hope that our work will contribute to the development of new therapeutic strategies for managing this challenging condition.

Minor points

7. Reading of the paper is very difficult and the abbreviations used increase the difficulty. For example, the peripheral vein is abbreviated as PIV, why?

We have made revisions to the text by replacing the abbreviations with the full name of the sample types. Specifically, "PIV" was replaced with "peripheral vein," "HV" was replaced with "hepatic vein," and "BDc" was replaced with "before discharge" in the main text.

Reviewer #2 (Remarks to the Author):

This interesting paper considers metabolite changes, measured by mass spectrometry techniques in hepatic venous blood and peripheral venous blood, as a result of transjugular intrahepatic portosystemic shunt (TIPS). Mass spectrometry analyses suggested decreases in bile acids in peripheral venous blood post TIPS vs pre TIPS. Further, those patients who subsequently developed a more severe hepatic encephalopathy (HE) after TIPS showed a greater decrease in bile acids and glycerophosphocholine levels post TIPS vs pre TIPS in peripheral venous blood samples. A suggested interpretation of these findings was that intrahepatic shunting may be occurring in patients who develop HE post-TIPS. Conversely, those patients with established intrahepatic shunting pre-TIPS were less likely to develop HE post-TIPS.

We sincerely appreciate the second reviewer's thoughtful evaluation of our paper, highlighting the intriguing findings regarding metabolite changes in hepatic venous and peripheral venous blood samples following TIPS. Their insightful comments have helped us improve our manuscript considerably."

1. Mass spectrometry analyses of hepatic venous blood and peripheral venous blood pre- and post-TIPS provide a novel snapshot of metabolite levels at two specific time points. As the authors comment, their assumption that mass spectrometry measurable metabolites are the toxins responsible for inducing HE post-TIPS may not be valid.

We would like to thank the reviewer for bringing up this important point. We agree that it is possible (and likely) that other toxins not measured by the specific mass spectrometry methodology utilized could also be responsible for inducing HE. We have modified the discussion to clarify this point:

"Participants who did not develop post-TIPS HE did not demonstrate the decline in bile acids (Figure 3C & Figure 4)... Of note, the methodology used here provides a snapshot of metabolites possibly implicated in HE. It is possible that toxins other than those captured by our method contribute to cognitive dysfunction in HE."

2. The authors have interpreted their observation of surprisingly small differences between the pre- and post-hepatic vein mass spectrometry metabolome as suggesting considerable intrahepatic shunting was already occurring in most participants prior to undergoing TIPS. Have the authors considered the impact of surgery as a confounding factor on the hepatic vein metabolome, which may be masking changes due to TIPS?

The reviewer raises a concern regarding the potential effects of surgery on the hepatic vein metabolome. The collection of hepatic vein blood before and after TIPS happens while the patient is under anesthesia. Thus, since for each participant samples were collected under similar conditions and within a short time frame from each other, we concluded that this variable did not differ between our pre- and post-TIPS hepatic vein

samples. For this reason, we believe most differences in metabolome pre- to post-TIPS are the result of the TIPS procedure itself.

We have modified the methodology to clarify that hepatic samples were collected under similar conditions and to include the time difference for collection of pre- and post-TIPS samples:

“Hepatic vein samples were collected under similar conditions during the TIPS procedure while patients were under anesthesia. After the procedure but prior to admission... Overall, the time difference between collection of samples was: less than an hour for hepatic vein samples collected before and after TIPS; 1-3 hours for peripheral samples collected before and after TIPS; and 14-19 hours for peripheral samples collected after TIPS and before discharge.”

3. Appreciating that efforts to characterise intrahepatic shunting have previously been sparse, did the authors consider an independent measurement of this phenomenon in their cohort pre-TIPS? Additional evidence of the presence or absence of intrahepatic shunting pre-TIPS would strengthen the interpretation that intrahepatic shunting may be occurring in patients who develop HE post-TIPS.

We thank the reviewer for this comment. We agree that having an independent measure of intrahepatic shunting pre-TIPS would strengthen our final conclusion that the presence of intrahepatic shunting pre-TIPS may be protective against the development of HE. However, these findings were surprising and unexpected and so having additional shunting measurements was not a part of our study design. We describe more reasons for why we did not measure intrahepatic shunting in our response to Reviewer 1, Comment 1 above. To address this comment, we have made the following modifications:

1. In the Results, we replaced figure 2b based on new analysis and results (**Reviewer Figure 2**) which used an alternative way to calculate the pressure/gradient based on HE grade (further explained in our response to reviewer #1 comment 3.3). The new results are consistent with our previous findings.
 2. In the Discussion, we have made modifications to the text as described in response to reviewer#1 comment 1.
4. The Abstract focuses on the background to the study rather than describing the patient cohort, the techniques used or discussing the hypothesis of less hepatic shunting pre-TIPS as an indicator development of HE post-TIPS

We thank the reviewer for this comment. We have modified our abstract to include the information on our study cohort, the techniques we use, our hypothesis, and our surprising finding relating to less intrahepatic shunting pre-TIPS in individuals who develop HE. Our modified abstract is found below:

“Elective transjugular intrahepatic portosystemic shunt (TIPS) placement can worsen cognitive dysfunction in hepatic encephalopathy (HE) patients due to toxins, including possible microbial metabolites, entering the systemic circulation. We conducted untargeted metabolomics on a prospective cohort of 22 cirrhotic patients undergoing elective TIPS placement and followed up to one year post TIPS for HE development. Our results suggest that pre-existing intrahepatic shunting predicts HE severity post-TIPS. Bile acid levels decreased in the peripheral vein post-TIPS, and the abundance of three specific conjugated di- and tri-hydroxylated bile acids were inversely correlated with HE grade. Bilirubins and glycerophosphocholines underwent chemical modifications pre to post TIPS and based on HE grade. Our results suggest that TIPS-induced metabolome changes could impact HE development, and that pre-existing intrahepatic shunting could be used to predict HE severity post-TIPS.”

5. The patient cohort (Table S1) comprised of eight female and 14 male subjects. Were any sex/gender based sub-analyses of the mass spectrometry findings undertaken? As this was a multi-centre study, were any differences noted in the mass spectrometry findings pre-TIPS between the two centres?

We appreciate the reviewer for this comment. We randomized the order and processed all plasma samples as one main batch for the mass spectrometry experiment in order to minimize technical differences between samples from different participants and facilities. In addition, we have investigated sex/gender and center as possible confounding factors, and present these results below:

- a. To determine if there were any sex/gender based differences, we ran an RPCA on either the hepatic vein or peripheral vein samples. Based on this analysis (**Rev. Fig. 4A-B**), we observed no sex/gender based differences in our metabolome. This was confirmed by running a PERMANOVA on the data which showed no significant sex/gender differences in either the peripheral or hepatic vein.
- b. To determine if there were any pre-TIPS differences we couldn't account for between the two facilities, we performed a RPCA analysis on the pre-TIPS peripheral or hepatic vein samples. The results from this analysis showed there was no significant difference between the two facilities pre-TIPS in the peripheral vein (PERMANOVA $P=0.125$, **Rev. Fig.4C**). However, there is a significant difference between pre-TIPS hepatic vein metabolomes when considering the two facilities (PERMANOVA $P=0.001$, **Rev. Fig. 4D**). We observed this pre-TIPS hepatic vein difference was driven by the $n=3$ samples from facility b. To determine whether these three points had an outsized effect on our results, we removed these three points from our hepatic vein pre- to post-TIPS dissimilarity analysis and observed that their removal did not affect our overall conclusion that there is a significant difference pre vs. post-TIPS based on HE grade (Kruskal-Wallis $p=0.013$) (**Rev. Fig. 5**). Likewise, a pairwise t-test also showed there is a significant difference between grades 0 vs. 2+ and 0 vs.1 (**Rev. Fig. 5**), similar to

what we observed in our original analysis in Fig. 2D. However, since our initial power calculation showed we needed at least $n=20$ participants and the removal of these points did not affect our overall results and conclusions from the hepatic vein, and for the sake of thoroughness in presenting our results, we believe these points should be included in the study.

Reviewer Figure 4. RPCA analysis based on sex/gender and center. Samples diversity based on sex/gender based differences in the (A) peripheral and (B) hepatic vein. Samples differences in the pre-TIPS metabolome based on center in the (C) peripheral and (B) hepatic vein. The triangle is the one serum sample.

Reviewer Figure 5. Hepatic vein metabolome dissimilarity based on HE grade removing (n=3) facility b samples. Group significance is based on a Kruskal-Wallis test while pairwise comparison significance is from a two sample t-test.

Minor points:

6. The terminology of pre-/post-HV, pre-/post PIV and BDC would benefit from being clearer.

We modified the text and replaced these abbreviations with the full name of the sample types. PIV was replaced with peripheral vein, HV was replaced with hepatic vein, and BDC was replaced with before discharge in the main text.

7. Some of the references are incomplete.

We would like to thank you for pointing out this error. We have modified the citations and bibliography to have complete references.

8. What was the time between blood sampling and sample centrifugation? Centrifugation occurred within ***

Blood sampling and sample centrifugation for collection of plasma fractions varied between 10 min to 2 hours. This could account for some of the between-site differences in the data. We have included this information in the methodology section.

9. Table S4 should be mentioned in the text.

Table S4 is mentioned in the results section of the text in the following way:

“A closer examination of the bile acid metabolome showed that the abundances of three conjugated di- and tri- hydroxylated bile acids in the post-PIV are significantly correlated with HE grade. All three decrease with higher HE grade, suggesting that circulating levels of these bile acids post-shunt placement are inversely associated with HE severity (Figure 4A, Table S4).”

REVIEWERS' COMMENTS

Reviewer #1 (Remarks to the Author):

The study was not purposely designed and therefore the interpretation of the results remains uncertain. The reviewed version essentially admitted the methodological limitations which in my opinion cannot be solved.

Reviewer #2 (Remarks to the Author):

Thank you for the detailed replies to the reviewers' comments. As mentioned in the author replies, the small difference in the blood metabolome of hepatic vein blood pre- and post-TIPS was an unexpected finding, and I suggest the serendipitous nature of this finding is emphasized in the Results, in addition to the Discussion.

Mention of the both the technique used for metabolomics (mass spectrometry) and the samples studied (blood) should be included in the Title and Abstract. Indeed, throughout the text the word 'blood' hasn't generally been included when mentioning peripheral vein or hepatic vein.

How effective was chatGPT in improving the manuscript? What proportion of the manuscript was edited by chatGPT?

REVIEWERS' COMMENTS

Reviewer #1 (Remarks to the Author):

The study was not purposely designed and therefore the interpretation of the results remains uncertain. The reviewed version essentially admitted the methodological limitations which in my opinion cannot be solved.

Though we are unable to answer the one specific question regarding differences between portal and systemic blood that the reviewer raised, we have demonstrated that we can answer questions regarding the relationship between pre- and post-shunt placement metabolomics and hepatic encephalopathy. We believe that this is of much greater interest to the community and clinically important. Moreover, our study provides evidence that the differences between portal blood and hepatic vein blood prior to shunt placement are minimal, which is transformational. This evidence provides the basis for exploration by future investigations. Future studies should be designed to test the hypothesis that intrahepatic shunting contributes to HE, which will further shed light into the possible mechanisms through which TIPS increases the risk of HE, such as portal toxins and microbially-derived metabolites. This paper provides the rationale and valuable initial data which will affect the design of these studies. As such, this study is valuable contribution to the field and transformational to our understanding of portal hypertension and hepatic encephalopathy.

Reviewer #2 (Remarks to the Author):

Thank you for the detailed replies to the reviewers' comments. As mentioned in the author replies, the small difference in the blood metabolome of hepatic vein blood pre- and post-TIPS was an unexpected finding, and I suggest the serendipitous nature of this finding is emphasized in the Results, in addition to the Discussion.

We would like to thank the reviewer for this suggestion. We have edited the manuscript to include the following text to the Results and Discussion, respectively, to emphasize this point:

“Serendipitously, within-participant dissimilarities between pre- and post-hepatic vein metabolome are significantly different between participants with an HE grade of 0 versus participants with a grade of 1 or 2+ (Wilcoxon $P=0.027$ and $P=0.036$, respectively, Fig. 2d, Data Table 8), despite no overall difference in hepatic vein blood metabolome before and after TIPS (Fig. 1b).”

“The surprisingly small differences in hepatic vein blood metabolome before and after TIPS suggests that the overall blood metabolome does not drastically change as a result of TIPS.”

Mention of the both the technique used for metabolomics (mass spectrometry) and the samples studied (blood) should be included in the Title and Abstract. Indeed, throughout the text the word 'blood' hasn't generally been included when mentioning peripheral vein or hepatic vein.

We have modified the title to include suggestions from the reviewer: “Portosystemic Shunt Placement Reveals **Blood** Signatures for the Development of Hepatic Encephalopathy **through Mass Spectrometry**”

How effective was chatGPT in improving the manuscript? What proportion of the manuscript was edited by chatGPT?

In response to reviewer's 1 comment that it was difficult to read the paper, we used chatGPT to clarify and make more accessible the section where we describe ChemProp, and how it can be used to better understand the biology related to the analysis. We used chatGPT as an editing tool as opposed to original content generation.